



# Presentation and discussion of the high resolution atmosphere-land surface-subsurface simulation dataset of the virtual Neckar catchment for the period 2007-2015

Bernd Schalge[1], Gabriele Baroni[2], Barbara Haese[3], Daniel Erdal[4], Gernot Geppert[5], Pablo Saavedra[1], Vincent Haefliger[1], Harry Vereecken[6,7], Sabine Attinger[8,9], Harald Kunstmann[3,10], Olaf A. Cirpka[4], Felix Ament[11], Stefan Kollet[6,7], Insa Neuweiler[12], Harrie-Jan Hendricks Franssen[6,7], Clemens Simmer[1,7]

[1]Institute for Geosciences, University of Bonn, Bonn, Germany

[2] Department of Agricultural and Food Sciences, University of Bologna, Bologna, Italy

[3]Institute of Geography, University of Augsburg, Augsburg, Germany

[4]Center for Applied Geoscience, University of Tübingen, Tübingen, Germany

[5]Department of Meteorology, University of Reading, Reading, England

[6]Forschungszentrum Jülich GmbH, Agrosphere (IBG-3), Jülich, Germany

[7]Centre for High-Performance Scientific Computing (HPSC-TerrSys), Geoverbund ABC/J, Jülich, Germany

[8]Institute of Earth and Environmental Science, University of Potsdam, Potsdam, Germany

[9]Helmholtz-Center for Environmental Research, Leipzig, Germany

[10]Institute of Meteorology and Climate Research (IMK-IFU), Karlsruhe Institute of Technology (KIT), Garmish-Partenkirchen, Germany

[11]Meteorological Institute, University of Hamburg, Hamburg, Germany

[12]Hannover, Institut für Strömungsmechanik und Umweltphysik im Bauwesen, Leibniz Universität Hannover, Germany

*Correspondence to*: Bernd Schalge (bschalge@uni-bonn.de)

**Abstract.**

Coupled numerical models, which simulate water and energy fluxes in the subsurface-land surface-atmosphere system in a physically consistent way are a prerequisite for the analysis and a better understanding of heat and matter exchange fluxes at compartmental boundaries and interdependencies of states across these boundaries. Complete state evolutions generated by such models may be regarded as a proxy of the real world, provided they are run at sufficiently high resolution and incorporate the most important processes. Such a virtual reality can be used to test hypotheses on the functioning of the coupled terrestrial system. Coupled simulation systems, however, face severe problems caused by the vastly different scales of the processes acting in and between the compartments of the terrestrial system, which also hinders comprehensive tests of their realism. We used the Terrestrial Systems Modeling Platform TerrSysMP, which couples the meteorological model COSMO, the land-surface model CLM, and the subsurface model ParFlow, to generate a virtual catchment for a regional terrestrial system



mimicking the Neckar catchment in southwest Germany. Simulations for this catchment are made for the period 2007-2015, and at a spatial resolution of 400m for the land surface and subsurface and 1.1km for the atmosphere. Among a discussion of modelling challenges, the model performance is evaluated based on real observations covering several variables of the water cycle. We find that the simulated (virtual) catchment behaves in many aspects quite close to observations of the real Neckar catchment, e.g. concerning atmospheric boundary-layer height, precipitation, and runoff. But also discrepancies become apparent, both in the ability of the model to correctly simulate some processes which still need improvement such as overland flow, and in the realism of some observation operators like the satellite based soil moisture sensors. The whole raw dataset is available for interested users. The dataset described here is available via the CERA database (Schalge et al, 2020): https://doi.org/10.26050/WDCC/Neckar_VCS_v1

## 1    Introduction

Earth environmental models are becoming increasingly important for climate and weather prediction, flood forecasting, water resources management, agriculture, and water quality control (e.g. Shrestha et al. 2014; Larsen et al. 2014; Simmer et al., 2015). Assuming that that the models are able to resemble the real-world based on state-of-the-art understanding of the system processes, the models are also used as "virtual realities" for hypothesis testing and decision support systems in many scientific disciplines (Clark et al., 2015, Semenova & Beven, 2015).

Virtual realities have been used for specific compartments of the terrestrial system in many studies (see Fatichi et al., 2016, and reference herein) and several advantages have been recognized. Bashford et al. (2002) computed virtual remote-sensing observations with 1 km resolution to derive, among others, process parameterizations for evapotranspiration in a hydrological model operating on the same scale as the remote sensing data. Weiler and McDonnel (2004) used a virtual-reality approach on the hill-slope scale to detect and quantify the major controls on subsurface flow processes and derive tunable parameters for conceptual models. Virtual experiments allowed Schlueter et al. (2012) to explore the relationship between soil architecture and hydraulic behavior and Chaney et al. (2015) to testing sampling designs. Hein et al. (2019) explores the relative importance of different factors in the hydrologic response of a catchment. Virtual realities are also often used to overcome limitations on the data-scarce observations. In this context, Ajami and Sharma (2018) used simulations results to test disaggregation method for soil moisture observations. In subsurface hydrology it is a standard procedure to test inverse modeling and data-assimilation approaches on virtual aquifers (e.g., Zimmermann et al., 1998; Hendricks Franssen et al., 2009), which are used to generate realistic aquifer data with exactly known hydraulic and geochemical properties at every point (e.g., Schaefer et al., 2002).

More recently, it has been highlighted that the terrestrial systems should be better exploited by the use of integrated models which are able to simulate water and energy fluxes in the subsurface-land surface-atmosphere system in a physically consistent way (Clark et al., 2015, Davison et al, 2018). For this reason, these integrated modeling approaches have also been considered to generate virtual realities (Mackay et al., 2015). However, despite the increasing computational capability and availability of





infrastructures, these modelling approaches are generally more technically demanding. In addition, the use of these types of
integrated models requires different expertises that are not usually cover within one single scientific group but requires strong
interdisciplinary collaborations among different partners. For these reasons, the use of these types of models is still not
commonly foreseen.
To overcome this limitation, in this paper we present the development, the testing and the data of a virtual reality of a mesoscale
catchment based on a fully integrated terrestrial model system. Our virtual catchment encompasses the terrestrial system from
the bedrock to the upper atmosphere covering the catchment of a higher-order river (length $\approx$ 380km, area $\approx$ 14000km$^2$)
including a buffer zone surrounding it, in which we simulate - as realistically as currently possible - the multi-year evolution
of states including the water and energy fluxes in and between all its compartments. We specifically venture to represent the
strong spatial variability of the land components, which affects the overall system behavior due to nonlinear couplings and
feedbacks. Since a virtual catchment with no resemblance to a real world catchment hardly allows for evaluating its realism,
we base our simulation loosely on the Neckar catchment in southwest Germany that contains quite variable topography,
different land cover, high and low precipitation regions, deep and low water tables and regions prone to flooding events. (see
Figure 1). The model does not aim at exactly reproducing the catchment's response to hydro-climatic forcing, instead we only
require that the simulated response is realistic with respect to typical spatial and temporal characteristics. For this reason, we
discuss the model realism in comparison with observations of the real catchment, but also its limitations, particularly in relation
to the chosen resolutions which balance the detail in process representation and computational feasibility
The remainder of the paper is structured as follows. In section 2, we introduce the simulation platform TerrSysMP, while
Section 3 describes in detail the surface and subsurface parameters for topography, soils and aquifers, land use, vegetation,
and the river network. In Section 4, we show snapshots and time series of state variables or system parameters extracted from
the virtual catchment and compare them to observations in the real Neckar catchment to demonstrate how well the most
important requirements are met. These results as well as possible ways to improve them are discussed in Section 5 together
with several issues, which came up during the development phase. We provide conclusions and an outlook in Section 6.

## 2    The Terrestrial Systems Modeling Platform (TerrSysMP)

We used the Terrestrial System Modeling Platform (TerrSysMP, see Shrestha et al. 2014; Gasper et al. 2014; Sulis et al. 2015)
developed within the Transregional Collaborative Research Centre TR32 (Simmer et al. 2015) for the generation of the virtual
catchment. TerrSysMP couples (Figure 2) the hydrologic flow model ParFlow v693 (Ashby and Falgout, 1996; Jones and
Woodward, 2001; Kollet and Maxwell, 2006), the land-surface model Community Land Model, CLM v3.5 (Oleson et al.,
2008), and the atmospheric model Consortium for Small Scale Modeling  (COSMO v4.21, Baldauf et al., 2011) via the Ocean
Atmosphere Sea Ice Coupling framework OASIS3 (e.g. Valcke et al., 2006), using a dynamical two-way approach including
down- and upscaling algorithms for fluxes and state variables between computational grids of different resolution.



ParFlow is a variably saturated watershed flow model, which solves the three-dimensional Richards equation to model
saturated and unsaturated flow in the subsurface, and the fully integrated kinematic wave equation to model two-dimensional
overland flow. Also other global and regional hydrological models use the latter to route overland flow, e.g. MODCOU
(Haefliger et al., 2015) and TRIP (Alkama et al., 2012). Advanced Newton-Krylov multigrid solvers are used that are especially
suitable for massively parallel computer environments. Excellent model performance and parallel efficiency have been
documented by Jones and Woodward (2001), Kollet and Maxwell (2006), and Kollet et al. (2010). A unique feature of ParFlow
is the use of an advanced octree data structure for rendering overlapping objects in 3-D space, which facilitates modeling
complex geology and heterogeneity as well as the representation of topography based on digital elevation models and
watershed boundaries.
CLM is a single column biogeophysical land-surface model released by the National Center for Atmospheric Research
(NCAR), which considers coupled snow, soil, and vegetation processes. Land surface heterogeneity is represented as a nested
sub-grid hierarchy in which grid cells are composed of multiple land units (glacier, lake, wetland, urban, and vegetation),
snow/soil columns (to capture variability in snow and soil states within each land unit), and Plant Functional Types (PFTs) to
capture the biogeophysical and biogeochemical differences between broad categories of plants in terms of their functional
characteristics. In TerrSysMP, the 1-D Richards-equation model included in CLM is replaced by ParFlow.
COSMO is a limited-area, non-hydrostatic numerical weather prediction model, which operationally runs at the German
weather service DWD, among others, for Numerical Weather Prediction (NWP) and various scientific applications on the
meso-β and meso-γ scale. COSMO is based on the primitive thermo-hydrodynamical equations describing compressible flow
in a moist atmosphere. As a limited-area model, COSMO needs lateral boundary conditions from a driving larger-scale model.
We impose the lateral conditions by nesting COSMO in COSMO-DE which spans Germany. At the lateral boundaries a
relaxation technique is used in which the internal model solution is nudged against an externally specified solution over a
narrow transition zone between the two domains.
Within OASIS3, the upscaling algorithm uses the mosaic or explicit sub-grid approach (Avissar and Pielke, 1989) in which
high-resolution land surface fluxes are averaged and transferred to the coarser resolution of the atmospheric model component.
The implemented Schomburg scheme (Schomburg et al., 2010, 2012) downscales atmospheric variables of the lowest
atmospheric model layer to the higher-resolved land surface model. The scheme involves (i) spline interpolation while
conserving mean and lateral gradients of the coarse field, (ii) deterministic downscaling rules to exploit empirical relationships
between atmospheric variables and surface variables, and (iii) the addition of high-resolution variability (i.e. noise) in order to
honor the non-deterministic part and to restore spatial variability.
TerrSysMP allows simulating the terrestrial water, energy, and biogeochemical cycles from the deeper subsurface including
groundwater (ParFlow) across the land-surface (CLM) into the atmosphere (COSMO). Water and energy cycles are coupled
via evaporation and plant transpiration; these processes are modeled by CLM with a non-linear coupling to ParFlow through
soil-water availability and root-water uptake (Figure 2). The two-way coupling between CLM and COSMO encompasses
radiation exchange and turbulent exchanges of moisture, energy, and momentum. OASIS3 allows for different temporal and



spatial resolutions of the coupled model components. For example, a temporal resolution of 15 minutes is sufficient for the
subsurface and land-surface components, whereas time steps as small as 5 seconds are needed for the atmosphere. A higher
spatial resolution can be assigned for the surface and subsurface parts to allow for a better representation of soil and land-use
heterogeneity.
Since high-resolution and long time-series of the fully coupled system are needed to satisfy our need to check the statistical
behavior of the system, the models were run on the IBM/BlueGeneQ System JUQUEEN at the Jülich Supercomputing Centre
(Jülich Supercomputing Centre, 2015). JUQUEEN has a total of 28672 nodes with 16 cores each. Our configuration involved
using 256 nodes for 12hours, restarting the simulation every 7 simulation days. This is necessary as the runtime for Parflow
can vary greatly depending on the conditions in the virtual catchment. The total number of grid cells for the domain is 323,675
per model layer with 10 layers for CLM and 50 layers for ParFlow, and 58,420 grid-points for the 50 COSMO layers resulting
in 22.3 million grid cells. We ran the fully-coupled model for a total period of nine years (2007-2015). On average the actual
runtime was approximately eight hours. This means that for one year of simulation about 1.7 million core-hours are needed.
For the full nine-year time-series that is about 12 million core-hours; another ~8 million hours were needed for the spin-up.
We used an output interval of 15 minutes, which results in a total output of 38.5TB of data for the full time-series, where about
half was produced by COSMO and a quarter each by CLM and ParFlow.

## 3   Description of the Virtual Catchment

Our virtual catchment is based on the Neckar catchment in southwestern Germany (see Figure 1), east of the Black Forest
mountain range and north of the Jurassic ridge of the Swabian Alb. The catchment has a varying topography including
mountains up to 1050 m a.s.l., river valleys, different land use types, i.e. grassland, cropland (majority of the area), broadleaf
and needle leaf forest (see Figure 3), and relatively large soil spatial variability. Annual mean precipitation over the real
catchment ranges between 500 and 2000mm (see Section 5.1) with highest values over the Black Forest. Inter-annual
variability of precipitation can reach up to one third of the mean value. Monthly precipitation can vary largely and its mean
annual cycle is weak with slightly lower values in spring and autumn. While summer precipitation is dominated by convection,
winter precipitation is predominantly related to fronts of extra-tropical cyclones with enhanced precipitation over the
mountains due to orographic lift. Daily average temperatures vary with altitude between -5°C and 0°C in January and between
13 and 18°C in July. Land use and cover in the lower elevations are dominated by agriculture while the Black Forest features
mainly needle-leaf trees. Broad-leaf trees can be found over smaller areas throughout the catchment. The distance to
groundwater is in large parts of the area restricted to a few meters, in particular in lowland areas, which assures strong coupling
between groundwater table and evapotranspiration (Maxwell et al., 2007). These typical central European catchment features
led us to choose the Neckar catchment as the basis for the virtual catchment.
The computational domain is a rectangular area of ~57,850km² encompassing the Neckar catchment of ~14,000km². The
domain is larger than the Neckar catchment in order to allow the atmospheric model to develop its own internal dynamics.



COSMO is run on a 1.1km horizonal grid with 230x254 grid points, which includes a 4 grid point-wide outer frame zone
where only the lateral boundary forcing is used without coupling to the CLM, as well as 50 vertical layers in hybrid coordinates
(terrain following at the surface, flat in the stratosphere). COSMO is set up identical to the operational COSMO-DE setup of
the German national weather service (Deutscher Wetterdienst, DWD), e.g., the deep convection parameterization is switched
off because at the chosen grid resolution convection is enabled by the dynamical core (see Section 2.1). In COSMO-DE, the
operational resolution is 2.8km, so that the approximation regarding deep convection is even more appropriate in our
simulations. Similar choices were taken by Smith et al. (2015), who simulated precipitation events of roughly the same domain
using nested WRF models, where the cumulus parameterization was switched off at horizontal resolutions of 900m and 300m.
Lateral boundary forcing and constant fields (topography, land-mask etc.) are provided by the COSMO-DE analysis fields,
which are downscaled to the 1.1km grid by linear interpolation. The lateral relaxation zone, which moderates the jump from
the lateral driving fields to the inner model area, is set to 12km.
A software restriction does not allow for cases with more than 4.2 million CLM columns; thus currently a higher spatial
resolution for CLM and ParFlow than 400 m cannot be used for the Neckar catchment. So, ParFlow and CLM use the same
horizontal grid with a resolution of 400 m and 535x605 grid points. The vertical grid for both component models is partially
the same, with CLM limited to 10 vertical layers up to a total depth of 3 m shared with ParFlow, which has in total 50 vertical
layers reaching down to 100m. COSMO runs with a 5sec timestep while CLM and Parflow run at 15min timesteps, which is
also the coupling frequency.
For setting up CLM, the European digital elevation model (DEM) by the European Environment Agency EEA
(http://www.eea.europa.eu/data-and-maps/data/eu-dem) was projected to the latitude/longitude grid and bi-linearly
interpolated to 400m from the original 30m spatial resolution. The same DEM is used to create the slope input files for ParFlow.
A slight modification to the original DEM was made in order to ensure that the simulated Neckar River would flow in the
correct valley, especially in the upper half of the catchment where the valley is not always properly resolved by the 400m
resolution. In total, the elevation of 8 grid points was reduced to achieve proper routing for the Neckar River. We have not
considered rivers outside the Neckar catchment in these corrections; thus there are cases where their routing is not identical to
the real rivers.
Land use is taken from the 2006 Corine Land Cover Data Set (http://www.eea.europa.eu/data-and-maps/data/corine-land-
cover-2006-raster-3) also provided by EEA. Since the latter dataset features many more land use types (at a resolution of
100m) than required by CLM, they were grouped according to the CLM (IGBP) Plant Functional Type classes (1) broad-leaf
forests, (2) needle-leaf forests, (3) grassland, (4) cropland, and (5) bare soil. Urban areas are not considered in this setup and
replaced by bare soil. Water surfaces (e.g., larger lakes like Lake Constance in the South of the domain) are also treated as
bare soil in CLM while COSMO uses its own land-mask and specific calculations for water surfaces. Therefore, no values
from CLM are used for water surfaces in COSMO. A few hundred grid cells feature shrubs (mostly areas that are re- or de-
forested or areas at higher altitudes) which are treated as forests, and each grid cell features only one – the most dominant –
plant functional type. The plant Leaf Area Index (LAI) is computed from MODIS (Myneni et al. 2002) as monthly averages



for the year 2008 for each of the four vegetated land use classes. This LAI is increased for all plant functional types by about
20 percent in the summer months and significantly changed from factors less than 1 to about 3.3 in winter-time for needle-leaf
forests in order to account for known biases in the MODIS data (Tian et al. 2004). The stem area index (SAI) is estimated from
the LAI by a slightly modified (no dead leaves for crops, constant base SAI of 10 percent of the maximum LAI) formulation
of Lawrence and Chase (2007) and Zeng et al. (2002) to better represent European tree types. Vegetation height was set to 7m
for needle-leaf trees and 10m for broad-leaf trees to account for partial coverage by shrubs, to 20 - 120cm for crops, and to 10
- 60 cm for grass depending on the time of the year with low values in the winter months and largest values in July and August.
Since we consider only one crop type, we do not specify a harvest date when the plant height drops to its minimum, but assume
a smooth decline between August and October.
For the representation of soils in CLM we use the 1:1,000,000 soil map (BUEK1000) provided by the Federal Institute for
Geosciences and Natural Resources - BGR
(http://www.bgr.bund.de/DE/Themen/Boden/Informationsgrundlagen/Bodenkundliche_Karten_Datenbanken/BUEK1000/bu
ek1000_node.html). This soil map is available for entire Germany; thus only small areas in Switzerland and France are missing
outside the Neckar catchment for which we assume a nearby soil class. BUEK1000 offers sand and clay percentages as well
as carbon content for two to seven soil horizons down to a maximum depth of 3m for each soil type. The carbon content is
used to infer soil color. For urban areas (modeled as bare soil, as mentioned above) a fixed soil color (class 8 in CLM) was
used.
Since soil properties may vary substantially at scales smaller than the 1km for which BUEK1000 is appropriate, which might
impact system dynamics (Binley et al. 1989, Herbst et al. 2006, Rawls 1983), the soil map is downscaled by artificially adding
variability using the conditional points method recently presented in Baroni et al. (2017) as follows:
(1)      The BUEK1000 soil map is randomly sampled at 1995 point locations with one sample every 5 km$^2$ on average, a
minimum sample distance of 250 m, and at least one sample for each soil type of the original soil map. This strategy resulted
from extensive testing by minimizing the tradeoffs between reproducing the main features of the original soil map and creating
variability at finer resolution.
(2)      The sample locations are used as conditional points for further interpolation. Here, texture, carbon content, and depth
of the first three soil horizons are extracted from the BUEK1000. In addition, the sand content of the original map was increased
by approximately 20% resulting in a slightly higher hydraulic conductivity because previous simulations yielded too shallow
unsaturated zones.
(3)      Experimental variograms and cross-variograms are calculated for all variables and exponential models were fitted to
all spatial structures.
(4)      A texture map (sand and clay percentage) is generated using a single realization based on conditional co-simulation
(Gomez-Hernandez and Journal, 1993) to provide the sub-scale variability (<1 km$^2$). Horizon depths and carbon content are,
however, assumed to have a smoothed spatial variability; therefore, they are interpolated based on ordinary kriging.





(5)     Since ParFlow describes retention and hydraulic conductivity curves based on van-Genuchten-Mualem parameters,
pedotransfer functions are applied to estimate these parameters. The pedotransfer functions of Cosby et al. (1984), Rawls
(1983) and Tóth et al. (2015) are used and selected based on data availability, applicability of the particular approaches, and
previous evaluations conducted in the area (Tietje and Hennings, 1996).
In order to keep soil porosity identical between CLM and ParFlow, we replaced the porosity calculation within CLM (which
uses a different pedotransfer function). The Manning's surface roughness was set to a constant value of $5.52 \times 10^{-4}$ h/m$^{1/3}$ and
the specific storage to $1 \times 10^{-3}$. The chosen surface roughness value results in a realistic base flow for the local rivers without
calibration. Repercussions of this choice are discussed in Chapter 6. Slopes of the main rivers are additionally smoothed to
avoid artificial ponded areas.
In order to allow for realistic flow in the saturated zone, the 3-D geologic model of the geological survey of the state of Baden-
Württemberg was used from which eleven rock types were defined for Baden-Württemberg (see Figure 4). Some characteristic
features of the domain, such as middle Triassic and Jurassic karst aquifers, are not included to avoid the manifold hydrological
challenges related to its modeling. For areas outside of Baden-Württemberg we extended the rock types at the boundary
outwards to cover the full computational domain. Tab. 1 summarizes porosity and hydraulic conductivity used in the domain
for the different stratigraphic units. As already mentioned above, karst features of limestones are not considered, and porosities
in stratigraphic units containing limestones and crystalline rocks are set considerably higher than in nature to somewhat counter
this.
Not covered by the discussed data sets are the large alluvial bodies filling large part of the Neckar valley throughout the domain
(Riva et al., 2006). Up to 30% of the runoff takes place in the subsurface especially during periods of base flow according to
a sub-catchment simulation performed for the year 2007. In that simulation we used measured precipitation and river discharge
data together with the simulated evapotranspiration to calculate the water balance over a whole year. While our simulated
evapotranspiration rates may be inaccurate, it is implausible that this can account for 30% of the precipitation; i.e. the water
could have left the domain only through the subsurface. Thus, gravel channels are needed to account for this lateral flow. Since
the valleys in the catchment are often small compared to the limited horizontal resolution of the model, we conceptualize the
alluvial bodies as gravel layers underneath all river cells (cells with a mean pressure head >0.1m) and directly next to rivers
(riverbanks, i.e., one grid point besides each river cell). The assumed gravel layers reach from beneath the soil down to a depth
of 8m. The gravel cells are parameterized with a high hydraulic conductivity of 1 m/h, a porosity of 0.6 and van-Genuchten
parameters of 2 for $n$ and 4m$^{-1}$ for $\alpha$ (residual saturation is 0.06 cm$^3$/cm$^3$). Our setup results in a reasonable distribution of
surface and subsurface discharge at the outlet of the catchment and reasonable river – aquifer exchange fluxes. In addition to
the gravel channels, we included a layer of weathered bedrock, which starts below the soil and extents down to a depth of 6m.
This layer is characterized by substantially larger porosity (0.4) and hydraulic conductivity (0.1 m/h) than the rock below. This
layer was added to enhance subsurface flow and counter the common occurrence of too shallow water levels if these features
are not included.





Since we enforce no-flow boundary conditions at the subsurface domain boundaries, all water has to eventually reach the
surface in order to leave the domain. This happens predominantly in areas outside of the Neckar catchment, e.g. in the upper
Rhine valley, thus soil-moisture values in this region may be too high.

## 4    Results

In the following, we present example results of the virtual-reality simulations in order to demonstrate its potential for a better
understanding of the dynamics in coupled terrestrial systems. We will also show that the simulations quite well resemble
observations in the real Neckar catchment, and thus can be used to develop and evaluate modelling and prediction strategies.
Precipitation is the strongest hydrological driver in this region; thus its realistic spatial and temporal variability in the domain
including its statistical relations with topography is important. Also, the state of the atmospheric boundary layer, which reflects
the interaction of the land surface with the atmosphere is a critical component of the terrestrial system, which should be
represented by the simulation with some confidence. Along with the comparisons we will also discuss the challenges
experienced with such a modeling setup.
Figure 5 shows as an example result a snapshot of the simulated three-dimensional distribution of cloud water/ice, precipitation
density, and volumetric soil moisture. The soil exhibits different soil moisture layers, the variability of which is mainly
connected to different soil hydraulic properties. Only clouds reaching high enough to have sufficient cloud ice produce
precipitation, and some precipitation evaporates before it reaches the ground. Extended weather fronts moving through the
domain (not shown), which are imposed by the boundary conditions, are also simulated realistically given the resolution of the
atmospheric model.

### 4.1    Relation between water table depth and evapotranspiration

An important measure for hydro-meteorological interactions within a catchment is the relation between water availability and
surface energy flux partitioning. Thus, the virtual catchment simulation should capture the expected reduced evapotranspiration
(ET) with increasing distance to groundwater (e.g., Maxwell et al., 2007; Shrestha et al., 2014). In Figure 6, we show daily
averaged ET (evaporation as all other contributors are zero) values over bare soil against distance to groundwater for 30[th] April
and 30[th] July for the year 2007. April was almost completely dry (on average less than 3 mm precipitation over the domain),
while July was much wetter, but the increased solar radiation and thus temperatures compared to April result in higher ET
rates and thus a quicker drying of the top layer of the soil. Figure 6 indicates a reduction in ET when the distance to groundwater
falls below 15 – 100cm, depending on soil properties with faster ET reduction for increasing soil sand contents. Such relations
are less obvious for cells with significant plant cover: while trees show overall higher ET and almost no ET change with
distance to groundwater due to their deep root zones, ET variability increases with larger distances to groundwater (not shown).
Also crops and grassland show limited ET changes as a function of distance to groundwater, which can, however, be explained
by the high water availability (no water stress) in the time period considered. Figure 6 also contains a small number of grid-





points at water table depth of 7m or deeper with evaporation rates only slightly lower than in the shallow water table regions.
These relate most likely to cells that retain high levels of upper-level soil moisture even during dry periods to support higher
evaporation. This could be either because of the soil properties or due to high flow from neighboring cells. Time-series of soil
moisture for the 10 days preceding 30[th] April and 30[th] July are shown in Figure 7 to illustrate this effect. It can be seen that
volumetric soil moisture for cells with deep water table but high evaporation is much more similar to cells with shallow water
table than to cells with deep water table but low evaporation. This means that these cells can retain high upper level soil
moisture despite having a low groundwater level. Most of these cells are near rivers and can receive just the right amount of
water from lateral flow to keep evaporation high while groundwater level stays low.

### 4.2  Precipitation

We compare the simulated precipitation with the 1 km$^2$ gridded REGNIE product of DWD, derived from in-situ precipitation
observations (Rauthe et al., 2013). For the evaluation of seasonal daily precipitation cycles hourly observations of 71 DWD
observational stations are used. The simulated seasonal mean precipitation (Figure 8) and the annual mean precipitation (not
shown) are governed by the orographic structures of the Black Forest and Swabian Alb. Values range between approximately
520 mm/year around Mannheim and 2105 mm/year over the Black Forest in good accordance with REGNIE concerning the
overall pattern and range (510 mm/year – 2130 mm/year). Overall the simulation shows about 10% higher annual precipitation
in the east and south and about 25% lower in the north and west compared to REGNIE. During winter (December to February)
precipitation is dominated by advection from the west, which result in maxima over the upwind and peak zones of the
mountains and leeward minima. The simulated winter pattern (j) compares well with REGNIE (k), but the model
underestimates precipitation in the northwestern part of the catchment (l). Over the mountains a slight lateral shift of this kind
of precipitation pattern results in neighboring areas with under- and overestimation also found for COSMO simulations coupled
to its own TERRA land surface model (e.g., Dierer et al., 2009; Lindau and Simmer, 2013). In fall, the difference pattern
between simulations and REGNIE (i) is similar to the winter pattern, but has smaller contrasts. In spring, the simulated
precipitation is higher compared to REGNIE. In the summer (June to August), cloud bases are usually higher and reduce the
patterns caused by the luff-lee effects. Moist air extends further to the east and south and gets staunched by the alpine upland
leading to enhanced precipitation there. The simulated summer precipitation pattern, which is dominated by convective
precipitation, resembles the REGNIE pattern but exceeds the latter by about 20% lower over large parts of the catchment
(Figure 8).
The mean seasonal diurnal precipitation cycles (Figure 9) reflect the dominating precipitation types. While observed and
simulated winter precipitation (Figure 9b) do not show a diurnal cycle, summer precipitation (Figure 9a) increases over the
afternoon reaching a maximum at about 7pm in accordance with the maximum of convective precipitation. The simulations
reproduce this pattern but exhibit a weak second peak between 6am and 12am while the afternoon/evening increase is delayed
by about two hours. The simulated daily precipitation distribution fits the observations quite well. While the virtual catchment
has somewhat less dry and low precipitation days than REGNIE, the number of days between 4 and 10 mm are higher than in





REGNIE (not shown). The simulated and observed seasonal precipitation cycles (Figure 10) compare very well and mean
precipitation is nearly identical between simulations and observations. The model reproduces the seasonal cycle of maximum
daily precipitation well, however with larger differences in the summer (see also Dierer et al. 2009).

**4.3     Atmospheric State Variables and Surface Radiation**

We compare the atmospheric boundary layer (ABL) of the virtual catchment to observations from the meteorological tower at
Karlsruhe Institute of Technology (KIT; Kalthoff and Vogel, 1992) and with DWD radiosonde observations in Stuttgart (STG)
(see Figure 3 for locations and Table A1 for details of observed quantities). To avoid a biased comparison due to land-cover
mismatches between the simulation and the actual land use at the observation sites, the simulations are averaged over five-by-
five atmospheric grid boxes centered around the observation sites.
The 10m mean diurnal minimum temperatures in the virtual catchment are between 0.5 K (January) and 2.5 K (August) higher
than observed (Figure 11, top) and are reached approximately one hour later than observed with the subsequent morning
temperature rise shifted accordingly. The simulated diurnal temperature maxima are on average 0.7 K lower than in the
observations and are reached 30 minutes later than measured. The morning temperature gradient in the simulation ranges from
0.10 K/h in December to 0.31 K/h in April, which compares reasonably well with the observations (0.13/0.52 K/h in
January/April). The evening cooling, however, progresses too slowly and results in too high minimum temperatures. At 100
m above ground, diurnal maximum temperatures agree within 0.7 K while the warm bias of diurnal minimum temperatures
(0.9 K) is smaller than at 10m height (Figure 11, bottom). Also at 100 m a 1h shift between the diurnal minimum temperatures
and the morning temperature rise are found. In 200 m height, the simulated monthly mean diurnal cycles are practically
identical to the KIT observations (not shown). The temperature standard deviations in the virtual catchment are somewhat
smaller than observed, in particular during the afternoon for the summer half year when the reduction can be more than 20%.
COSMO in TerrSysMP estimates ABL heights via the bulk Richardson number criterion with a threshold of 0.22 for unstable
and 0.33 for stable conditions (Szintai and Kaufmann, 2008). Both seasonal and diurnal variations of the mean ABL height at
0 and 12h local time agree well with the observations using the same criterion (Figure 12), but the simulation tends to
overestimate ABL heights at nighttime by up to 150 m and underestimate it at daytime by up to 200 m in March. Figure 13
compares simulated mean vertical profiles of temperature, virtual potential temperature, and specific humidity with radiosonde
observations at 0 h and 12 h local time in Stuttgart (STG) including the mean differences (bias) and the standard deviation of
the differences. Simulations are up to 0.9 K warmer close to the surface at 0 h and up to 0.5 K colder at 12 h. At larger heights,
the simulations are up to 0.5 K warmer depending on land cover. Specific humidity profiles at 0 h are approximately 0.2 g/kg
too dry close to the surface and 0.2 g/kg too wet above 1500 m. At 12 h profiles are up to 0.3 g/kg too wet throughout. The
simulations have smaller virtual potential temperature gradients and are thus less stable close to the surface at 0 h. At 12 h, the
decreasing virtual potential temperature close to the surface is not captured and tends towards a more neutral instead of unstable
profile at low heights.



At KIT (STG) the land surface receives on average 20 W/m$^2$ (5.3 W/m$^2$) more incoming shortwave radiation and 18 W/m$^2$ (8
W/m$^2$) less incoming longwave radiation indicating a somewhat lower cloud cover (or lower cloud optical depth) as observed.
At daytime (6 h – 22 h), the mean outgoing longwave radiation matches the KIT observations, while at nighttime (22 h – 6 h)
values are 7.2 W/m$^2$ larger than observed, which corresponds to a higher surface temperature of approximately 1.4 K.
Overall, the atmospheric profiles, including the ABL heights, compare very well with observations. Noteworthy differences
only occur close to the surface with too high nighttime temperatures (up to 2.5 K in summer) and subsequently too small
morning temperature gradients. Somewhat higher incoming shortwave and lower incoming longwave radiation at the surface
indicate less cloud cover (or lower cloud optical depths) compared to the observations. These results are in line with a previous
evaluation of a 2.2 km COSMO simulation (Ban et al. 2014). In addition, we note somewhat reduced unstable conditions at
daytime close to the surface in the simulations.
**4.4    Passive Microwave Observations**
The most direct area-covering observations of soil moisture are currently provided by L-Band (1.4 GHz) passive microwave
observations from satellites. The Community Microwave Emission Model (CMEM) is used as a forward operator to simulate
the brightness temperatures (TB) at this frequency in vertical and horizontal polarization (de Rosnay et al., 2009). CMEM
simulates brightness temperatures at the top of the atmosphere resulting from microwave emission and interaction by soil,
vegetation, and atmosphere based on the state variables of the virtual catchment. Input to CMEM are the percentages of clay
and sand in the soil, the coverage with open water surfaces, the profiles of soil moisture and soil temperature, vegetation types,
and leaf area index (LAI). Satellite orbit geometry, antenna pattern, foot-print and incidence angle are taken into account
following the ESA SMOS (Soil Moisture Ocean Salinity) instrument specifications, i.e. a full-width-half-maximum field of
view leading to a footprint of 40km across-orbit and 47km along-orbit at multiple incidence angles (Kerr 2001) is applied.
This antenna pattern weighs the grid-cell simulated brightness temperatures (Figure 14, left) in order to obtain virtual SMOS
observations. Finally, these synthetic observations are rendered according to pixels based on the Icosahedral Snyder Equal
Area (ISEA) projection at a spatial separation of about 15 km similar to the SMOS L1C TB data product (Figure 14, right),
which can then be compared with real observations for an indirect evaluation of the simulation. Every pixel corresponds to a
fixed geo-location of the real SMOS L1C data product over the modeled area. Optionally, the satellite observation operator in
TerrSysMP is able to also replicate the NASA SMAP (Soil Moisture Active Passive) radiometer (Saavedra et al., 2016) for
years beyond 2015 since when SMAP data is available.
We evaluate the simulated brightness temperature distribution over the domain with real SMOS observations between April
2011 and September 2011. The real SMOS observations are corrected from radio-frequency interference (RFI) effects over
the region following Saavedra et al. (2016). Initial results with CMEM adapted parameters for surface roughness and
vegetation optical thickness (which needed to be increased from its standard values found in the literature), lead to a systematic
underestimation of the brightness temperature of about -20K on average (see orange line in Figure 15, which compares real



SMOS observations with the simulated brightness temperatures) and maximum and minimum differences of -33K and -6K,
respectively, for an incidence angle of 30°. A similar underestimation of -14K resulted for the 40° incidence angle with
maximum and minimum values of -34K and +15K (lower plot in Figure 16). Those differences are mainly caused by the too
large near-surface soil moisture values in the virtual catchment. The cumulative distribution functions of the satellite-derived
soil moisture products and the soil moisture of the virtual catchment suggests an about 63% higher near-surface soil moisture
compared to the satellite estimates (Saavedra et al. 2016, Figure 6) with extremes of 44% and 95%. A daily matching of the
cumulative distribution functions of the virtual catchment and satellite retrieved soil moisture are performed to find a factor
which then is assumed to be the soil-moisture bias of the simulation. Figure 16 compares true SMOS observations with
simulated brightness temperatures obtained without and with day-to-day correction for the assumed soil-moisture bias of the
simulation. The correction decreases the average bias in brightness temperature from -20K(-14K) to about -3K (-2K) for the
incidence angle of 30° (40°) at horizontal polarization. Similar results are found when the simulations were statistically
compared with observations of later years from the NASA SMAP (Fig. 3 in Saavedra et al. 2016). The remaining bias can
probably be further reduced by fine tuning radiation interaction parameters in CMEM, and by including orographic effects on
the effective incidence angle.  These biases will be addressed by an improved exploitation of the uncertainty of the radiation
interaction parameters and by including in CMEM a two-stream approximation to better simulate cases with dense vegetation
in the future.
The microwave observations retrieved from the virtual catchment show a typical situation encountered in data assimilation;
more often than not there are biases between simulated and remote sensing observations. This discrepancy usually has multiple
causes, which can relate to the observations themselves, assumptions in the observation operator used to simulate the virtual
observations, and in the model used to generate the systems state variables entering the observation operator. Even if these
differences cannot be removed, such observations can be highly valuable for data assimilation as long as temporal tendencies
are meaningful information. Usually the bias is statistically corrected and thus only the information in the temporal and (if
meaningful) spatial variability of the observations is exploited for moving the model states towards the true states.
**4.5    Evaluation of River Discharge**
We compare river discharge in the virtual catchment with observations made in the Neckar catchment at the gaging stations
Rockenau, Lauffen, and Plochingen for a three-year period from 2007 to 2009 (Figure 16). The range of the hydrological
response to precipitation in the virtual catchment is in adequate agreement with the observations; this is noteworthy since no
calibration to runoff data has been applied to the model. The simulated discharge peaks are, however, higher and delayed by
about one to three days compared to the observations. A reason could be a too large Manning´s coefficient and the model
resolution. In the discussion we suggest a scaling of Mannings coefficient to account for the mismatch between true river width
and the model resolution in order to better represent realistic flood dynamics. In spring and summer, the response to
precipitation is significantly smoother than observed and peak amplitudes vary with respect to peak amplitudes of the





observations. The differences between observed and simulated precipitation discussed above and the effects of the less
predictable convective events during these seasons may also play a significant role. Convective events will always be displaced
in space and time compared to the observations and may even show different individual life cycles including lifetime and
amplitude. Finally, the base flow is much lower compared to the real catchment during dry periods, most likely because the
grid resolution is considerably larger than the actual river width and the unresolved subsurface spatial heterogeneity. An
increased hydraulic conductivity via an increased soil sand content may reduce the base flow further as infiltration increases.
The results are further evaluated comparing the flow duration curve and the monthly run off coefficient. The former represents
the statistical probability to exceed a specific discharge value within a given time period while the latter is the ratio between
runoff and precipitation over the catchment area. Figure 17 shows the lower exceedance probability of the virtual catchment
compared to the observations, in particular for low discharge rates, a behavior attributed to the lower base flow component
and confirmed by the too low runoff coefficients in spring and summer but similar coefficients during the rest of the year
(Figure 18). We hypothesize that in this period the simulation has a lower hydrological response also due to missing subsurface
heterogeneity. As stated above, we have neglected karst features, which are known to produce fast lateral subsurface flows.
Overall, the model captures the general statistical features of the catchment including the typical seasonal trends quite well,
while differences are noted related to hydrological extremes and base flow. These differences could be reduced by model
calibration from which we refrain because hydrological extremes are not primary the objective of this study. We discuss
options to improve the representation of river discharge further below.

### 4.6    Groundwater

A plausibility check of the groundwater levels is performed in two steps. First we visually inspect the groundwater depth map,
shown in Figure 19a. Accordingly, the model shows a good distribution between shallower and deeper (5 meter and below)
groundwater tables. Furthermore, the deeper sections are found in the mountainous areas of the model domain, which
corresponds well with the real situation. It has to be noted though that regions with shallow groundwater levels often show
very small values, likely not to be found in the real catchment where the unsaturated zone is usually thicker. In a second step,
we compare simulated hydraulic heads with available data. The environmental protection agency of the state of Baden-
Württemberg (Landesanstalt für Umwelt, Messungen und Naturschutz – LUBW) operates 33 continuous groundwater
observation wells. Comparing those point measurements to simulation results of an uncalibrated model with 400m grid
resolution makes little sense. Instead, we compare (1) the magnitude of the fluctuation in the groundwater table throughout the
catchment during a year (calculated as the groundwater observation minus its mean, shown in Figure 19b) and (2) the average
trend of the groundwater in the full model (calculated after subtracting the mean and scaling the fluctuations to have the same
magnitude. According to Figure 19c, the magnitude of the groundwater fluctuations are within similar ranges as the
observations (Figure 19b), while a few real observation wells show larger magnitudes. Also the trends overall follow similar
patterns (Figure 19c). Hence, the groundwater, given the coarse resolution of the model in comparison with the compared point
measurements, shows a reasonable behavior.




## 5    Discussion

The size of the catchment and resolution considered (400 m) pose an enormous challenge in terms of required CPU-time. Still, the applicability of Darcy's law with laboratory-based parameters can be debated as we have to resort to apparent model parameters to produce realistic mass fluxes in the compartments. By compromising these technical and physical aspects in the setup of the virtual catchment, we experienced several challenges; four of them will be discussed which we believe to be inherent to simulating energy and mass fluxes across compartment boundaries with partial-differential-equation-based, high-resolution coupled models.

**Representation of rivers and surface roughness:** River flow in the ParFlow module of TerrSysMP is simulated by an overland flow module. Overland flow appears when hydraulic heads in the top cells are above the land surface. As there is no discrimination between overland flow and river flow, rivers in the simulation have the width of the grid resolution whereas the real rivers may be significantly narrower. Overland flow is represented in ParFlow with the kinematic wave approximation of the St. Venant equations with the surface friction parameterized by Manning's coefficient. Typical Manning's coefficients when assigned to e.g. to a 400 m grid cell while in fact the river is much narrower, would result in too high discharge values during rain events and far too low ones during dry periods. In both cases the always too low water levels caused by the too wide rivers result in a poor representation of river-subsurface exchange. Our current choice of Manning's coefficient in ParFlow ($5.52 \times 10^{-4}$ h/m$^{1/3}$) results in realistic average discharge throughout the year, albeit at too low flow velocities. In order to compensate for this inconsistency, the Manning's coefficient could be scaled such that the overland flow velocity in river cells equals the river flow velocity as proposed by Schalge et al. (2019), which improves the phasing between simulated and observed discharge and the discharge peak. Similarly, the hydraulic conductivity of the model top layer for river cells could be scaled in order to reduce the loss of too much surface water to the subsurface caused by the too wide river cells. These issues will become even more severe when model resolutions are reduced, e.g., for ensemble-based data assimilation because of the even higher demands for computing efficiency.

**Coarsening of topography:** The still coarse topography of the virtual catchment reduces the true hill slopes where lateral flow on the surface and in the shallow subsurface takes place. This affects quick-flow components towards rivers. As shown by Shrestha et al. (2015), coarse topography directly impacts the storage of water in the unsaturated zone because drainage becomes less effective. This in turn can lead to an overestimation of latent and underestimation of sensible heat flux. Additionally, coarse-resolution model runs result in delayed and stretched discharge peaks in the rivers. The severity of this effect is proportional to the degree of topography smoothing, that is introduced by the coarser resolution; therefore, any change in subsurface parameters such as hydraulic conductivity will depend on the degree of coarsening and the location within a catchment. Especially in narrow valleys and in mountainous areas this will lead to an overestimation of soil moisture, which we have not yet compensated by changing other parameters. Recently a method has been proposed to improve these issues by scaling horizontal hydraulic conductivity (Foster and Maxwell, 2019).





**Soil parameters:** As outlined in section 2, the soil hydraulic parameters were generated based on soil maps of the real Neckar catchment. According to the maps, the soils in the catchment consist mainly of clay and silt, which have rather low saturated hydraulic conductivities and small air entry pressure values. In large areas of the domain, the water content in our first simulations was close to saturation, even for upper soil layers, and the infiltration velocities were unrealistically low. Reasons are the soil parameters, which do not capture the true soil heterogeneity; moreover, real infiltration often takes place in root channels, small fractures, and other small structures. Thus, infiltration is always underpredicted by models using observed soil parameters assuming homogeneity. Infiltration processes may be better captured with dual domain approaches, which are, however, computationally demanding. A workaround would be to change the soil hydraulic parameters in order to obtain stronger infiltration. Currently, we use an artificially increased sand percentage of the soils in order to stay consistent with the concept of the pedotransfer functions used in CLM. We will also test known scaling rules (e.g., Ghanbarian et al., 2015) to increase for example the saturated hydraulic conductivity for larger soil units. These rules should be applied on the soil hydraulic parameters, estimated by the pedotransfer functions.

# 6    Conclusions and Outlook

In the present study we show the development and the data generated based on a integrated subsurface-land surface-atmosphere system TSMP. Plausibility tests for the derived virtual reality which tries to mimic the Neckar catchment in southwestern Germany, show that the virtual catchment is able to reproduce realistic behavior when compared to measurements. Comparisons of simulated precipitation and ABL statistics show a very reasonable agreement with real observations. However, comparisons with observed passive microwave measurements by satellites shows clearly a systematic bias which is probably related to a mixture of systematic errors in the latter, assumptions in the used forward operator, parameterizations of land surface properties (soil parameters) in the simulation, and missing processes therein (e.g., preferential flow, hill-slope processes). The analysis also shows a realistic connection between evapotranspiration and distance to groundwater in the virtual catchment, while larger deviations from reality are found for river discharge dynamics. The deficiencies could be traced to the model resolution, which limits the often much smaller river widths to multiples of the model resolution, and to the way river discharge is handled in the ParFlow component of TerrSysMP. A new parameterization scheme proposed by Schalge et al. (2019) will avoid such problems in future model simulations. The main issues we face for the upper Neckar are too high soil moisture and shallow groundwater levels. Several ideas have been proposed to improve the setup including scaling of the surface roughness and soil parameters.

Overall the results are encouraging regarding the viability of the virtual reality as key input parameters to the land surface and subsurface show very good agreement with observations. For these reasons, the analysis show that the results can be used as a basis for the community for, among others, exploring feedbacks between compartments, identify in which conditions simplification of the models could be done (Baroni et al., 2019) or develop and test methods for assimilating observations across compartments . We encourage the scientific community to explore this data for the different applications. Within the



study we also highlighted some limitations mainly due to the still sever technical limitation and the IT-requirements. We anticipate however that more sophisticated versions of the virtual catchment (higher resolution, improved parameterization of sub-scale processes as discussed above) are already in progress that could be also compared to this virtual reality in further study.

## 7    Data Availability

The presented virtual catchment is available in the CERA database of the German Climate Computing Center (DKRZ: Deutsches Klimarechenzentrum GmbH) (Schalge et al., 2020) at https://doi.org/10.26050/WDCC/Neckar_VCS_v1. The full nine-year time series (2007-2015) for all three compartments has a size of roughly 40TB in compressed netCDF4 format. Nevertheless, we encourage the use of this data set for investigations on data assimilation, but also the general functioning of catchments including cross-environmental interactions and predictability studies can profit from such complete state evolutions of the regional Earth system.

The TerrSysMP model is built in a modular way and users are supposed to get the component models by themselves while the coupling interface is provided through a git repository (https://git.meteo.uni-bonn.de). As of now, registration is required to access the TerrSysMP git and wiki page.

Both ParFlow (https://parflow.org/) and CLM (http://www.cgd.ucar.edu/tss/clm/distribution/clm3.5/) are freely available for download from their respective websites or repositories. COSMO is not available, but the DWD supplies it free of charge for research purposes upon request. More information on this process can be found in the TerrSysMP wiki.

## Acknowledgements

This research is funded by the Deutsche Forschungsgemeinschaft (DFG, FOR2131: "Data Assimilation for Improved Characterization of Fluxes across Compartmental Interfaces"). The authors gratefully acknowledge the Gauss Centre for Supercomputing e.V. (www.gauss-centre.eu) for funding this project by providing computing time through the John von Neumann Institute for Computing (NIC) on the GCS Supercomputer JUQUEEN at Jülich Supercomputing Centre (JSC). We thank the members of HPSC-TerrSys (http://www.hpsc-terrsys.de/hpsc-terrsys/EN/Home/home_node.html) and Klaus Goergen in particular for invaluable technical support with the JUQUEEN supercomputer. Furthermore, we thank Prabhakar Shresta and Mauro Sulis for their preliminary work and introduction to the TerrSysMP modeling platform.We also acknowledge work done on an earlier version of this script by Jehan Rihani.

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





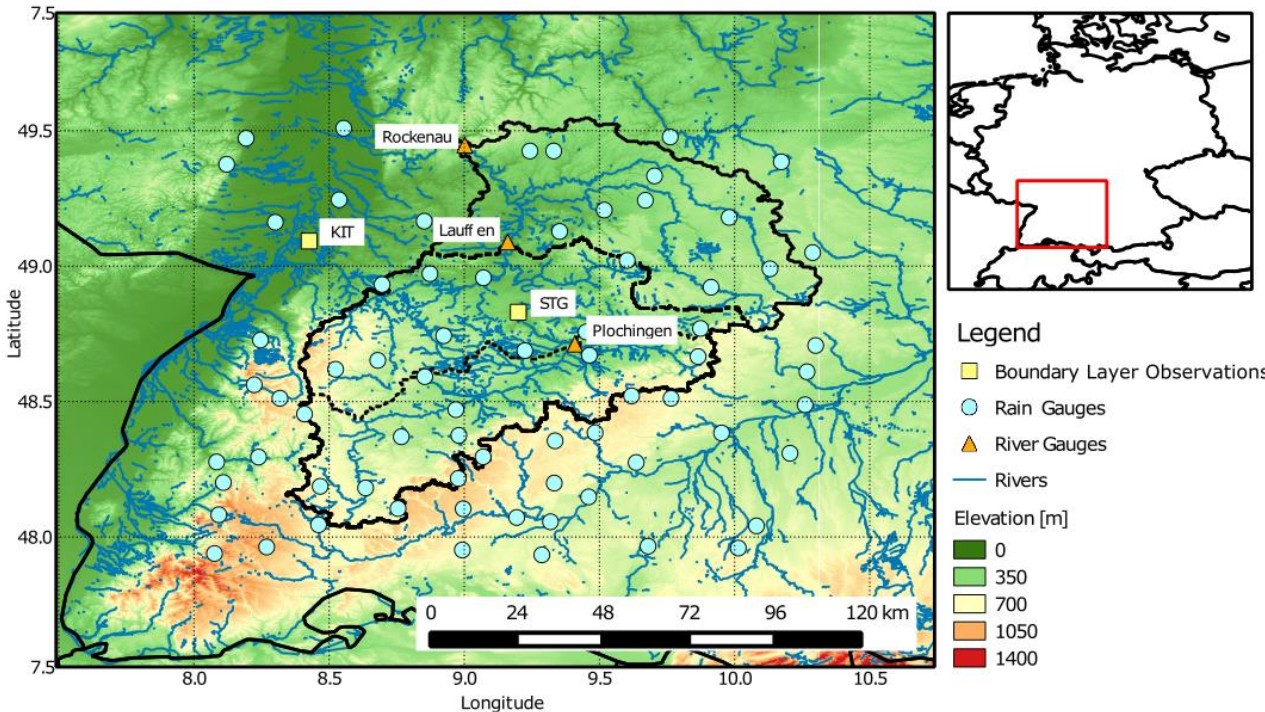


**Figure 1: Location of the Neckar catchment within SW Germany.**






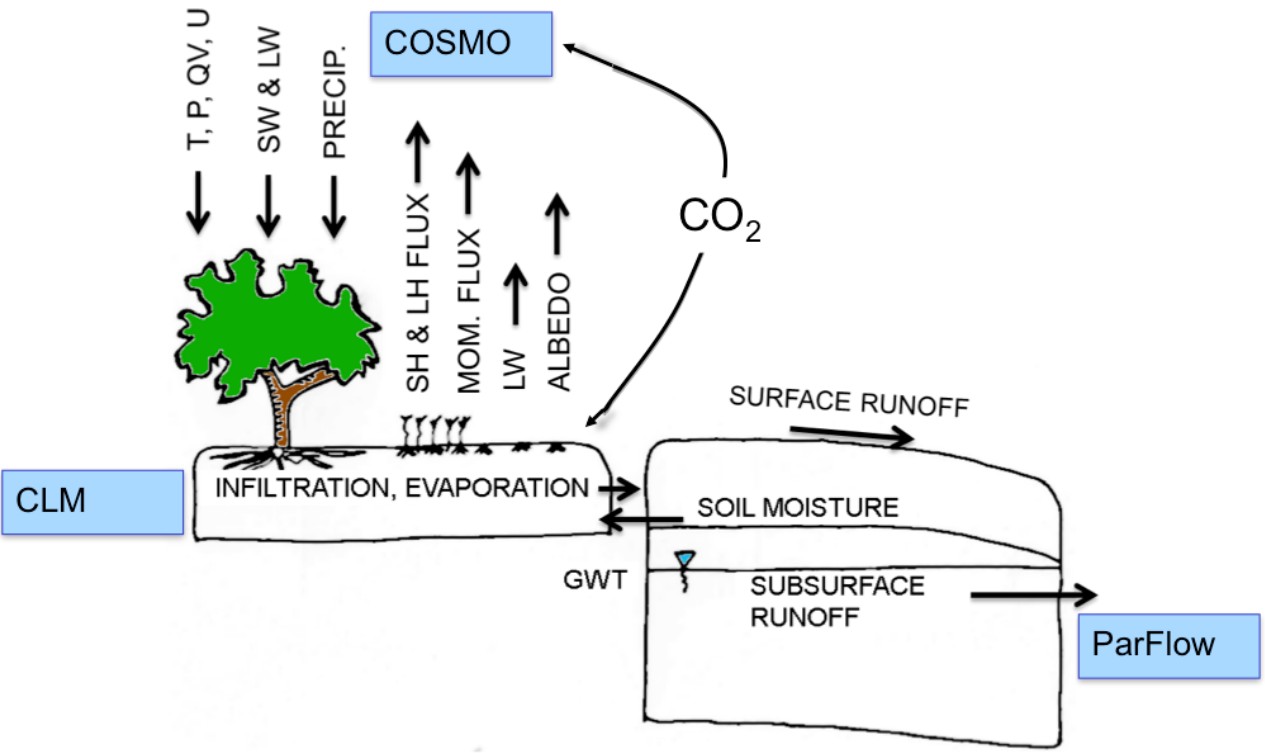


**Figure 2: Exchange of energy and mass fluxes in TerrSysMP (Gasper et al., 2014).**




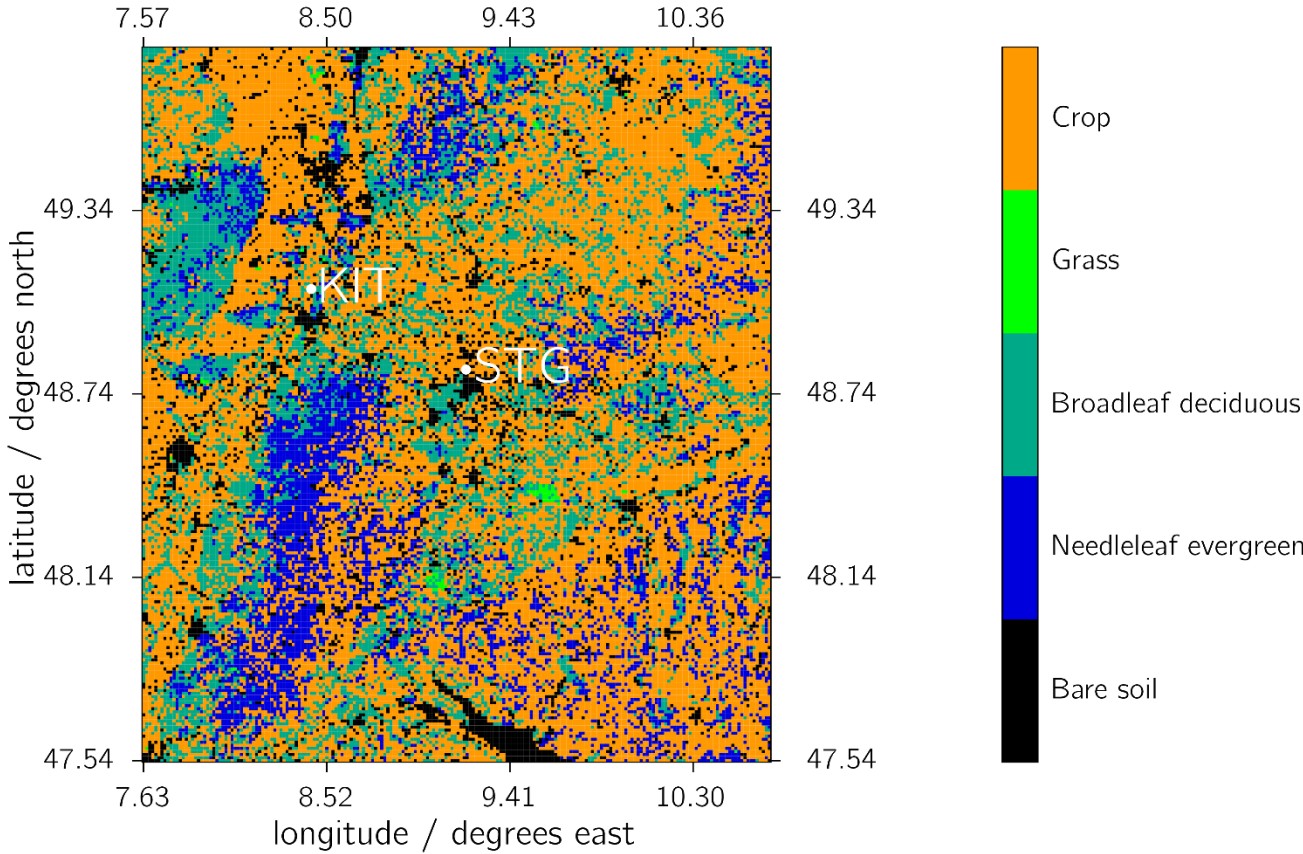


**Figure 3: Land cover in Domain 1 covering the entire Neckar catchment and bounding areas. KIT: Karlsruhe Institute of Technology (location of meteorological tower observations), STG: Stuttgart (location of radiosonde observations).**

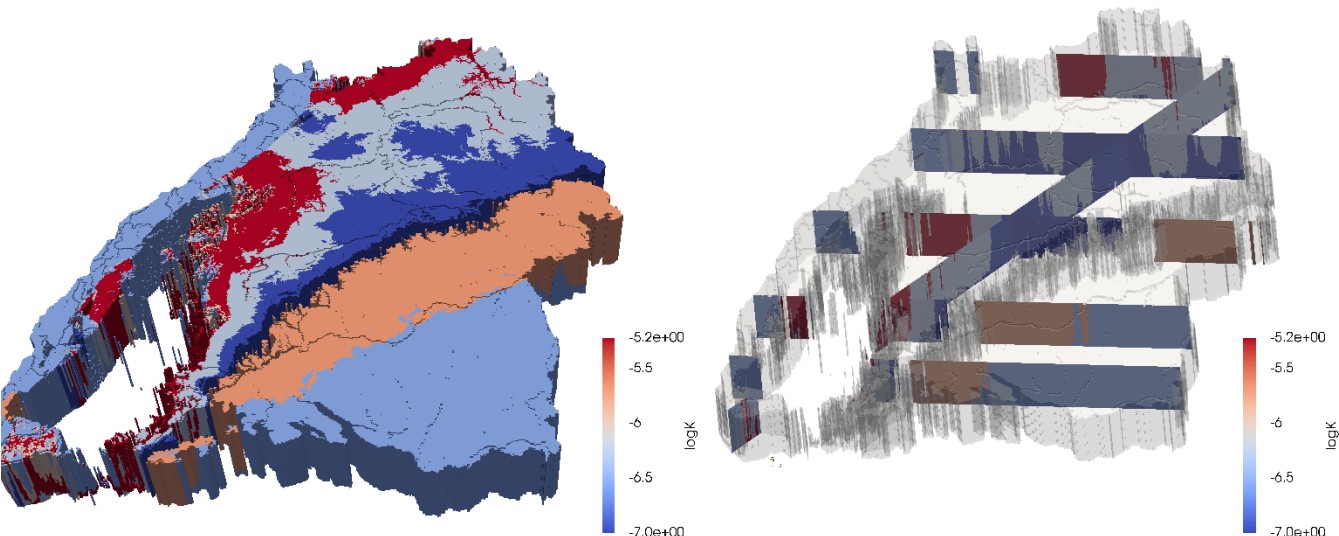

**Figure 4: Stratigraphy in the state of Baden-Württemberg represented by its logarithmic conductivity. The left figure shows a 3-D view of the 100 m deep geological model used in this work, where the elevation has been neglected for readability and the transparent regions corresponds to low-permeable material. The right figure shows the same using cross-sections to better visualize the vertical heterogeneity.**

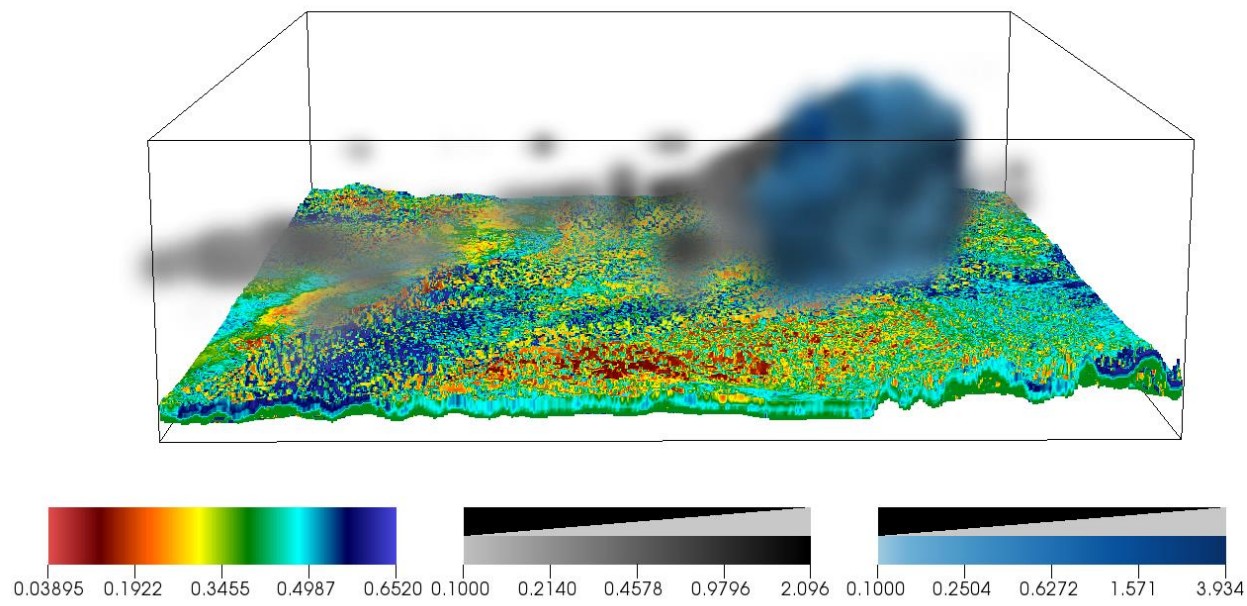

748

**Figure 5: Snapshot of the three dimensional distribution of cloud water/ice [g/kg] (greyscale), precipitation/rain water [g/kg] (blue in foreground over cloud) and soil moisture [cm3/cm3] (colored) at a time point with a single rain cloud with light rain.**



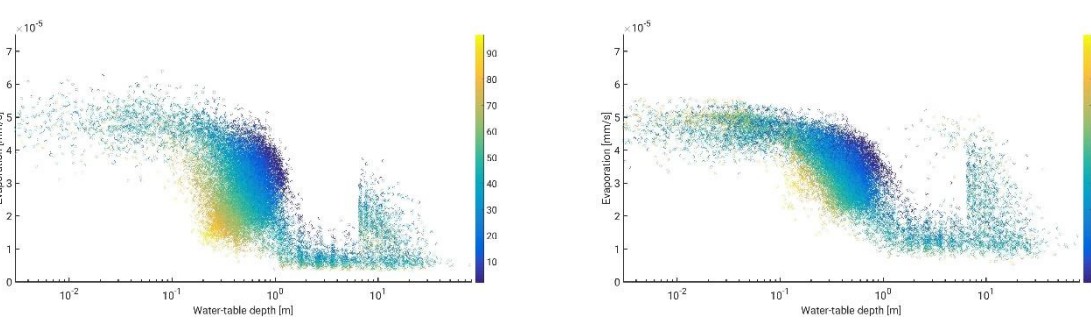

**Figure 6: Daily average evapotranspiration (ET) simulated for 30th April (left) and 30th July 2007 in [mm/day]. The color indicates soil sand percentage.**

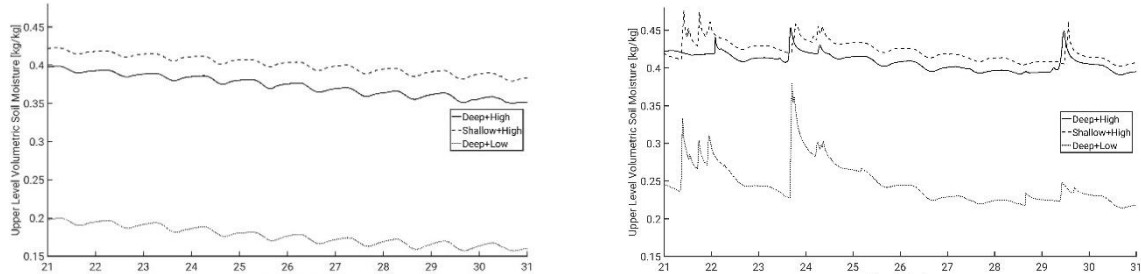

**Figure 7: 10-day time-series of volumetric soil moisture for three representative grid-points with high evaporation and shallow water-table (Shallow+High), low evaporation and deep water-table (Deep+Low), and high evaporation and deep water-table (Deep+High) prior to the days shown in Figure 6.**



**Figure 8: Mean seasonal precipitation over the Neckar catchment between 2007-2013 in the virtual reality (VR, left column) compared to the REGNIE data set (middle column). The difference between VR and REGNIE is shown in the right column. Figure (a), (b), and (c) show the comparison for spring (March – May); (d), (e), and (f) for summer (June – August); (g), (h), and (i) for fall (September – November); and (j), (k), and (l) for winter (December-February).**

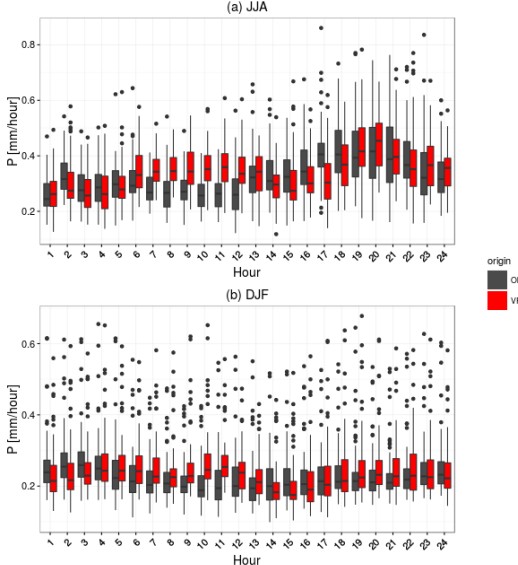

**Figure 9: Mean diurnal precipitation cycle for the 71 DWD stations and the corresponding simulations for wet days (more than 1 [mm/day]) for June-August (a) and December-February (b) season. The upper and lower hinges correspond to the first and third quartile, the center black line the median, the upper whisker (analog for lower whisker) extends from the hinge to the highest value within 1.5*(interquartile range), and the black dots mark the outliers.**

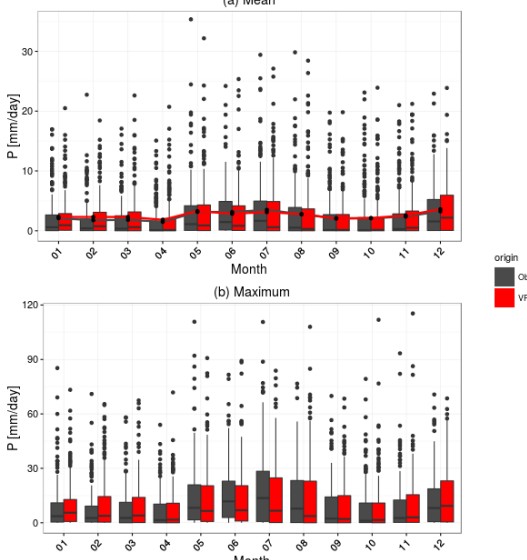

**Figure 10: (a) Daily precipitation distribution on a monthly basis as observed (black) and simulated (red). The gray and red lines indicate the monthly mean precipitation. (b) Maximum daily precipitation for the given months for the 71 DWD stations and the corresponding simulation. Box sizes as explained in the caption of Figure 10.**

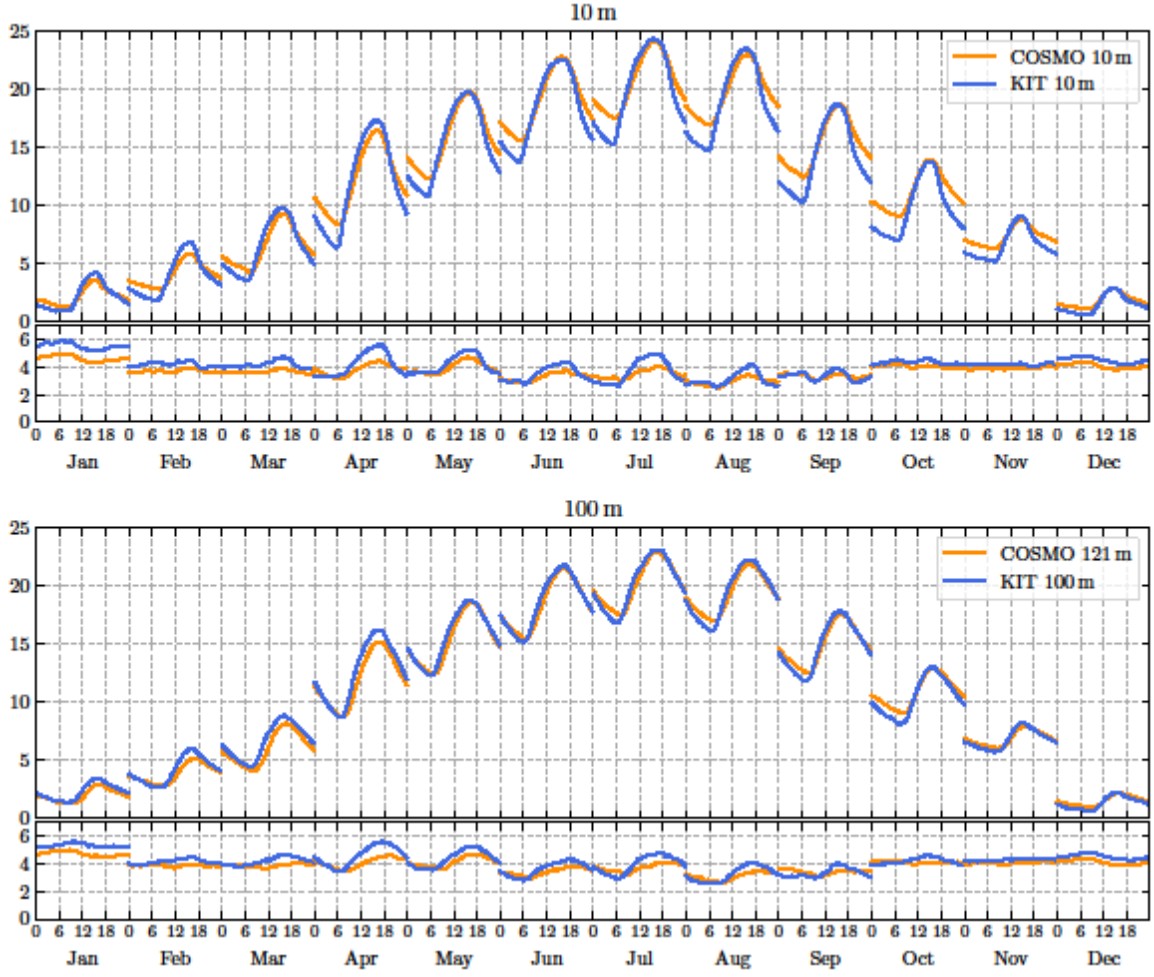


**Figure 11: Monthly mean diurnal cycles (local time) and respective standard deviation for air temperature in 10 m (top) and 100 m (bottom) height at the KIT tower and for the COSMO grid boxes around the KIT location.**




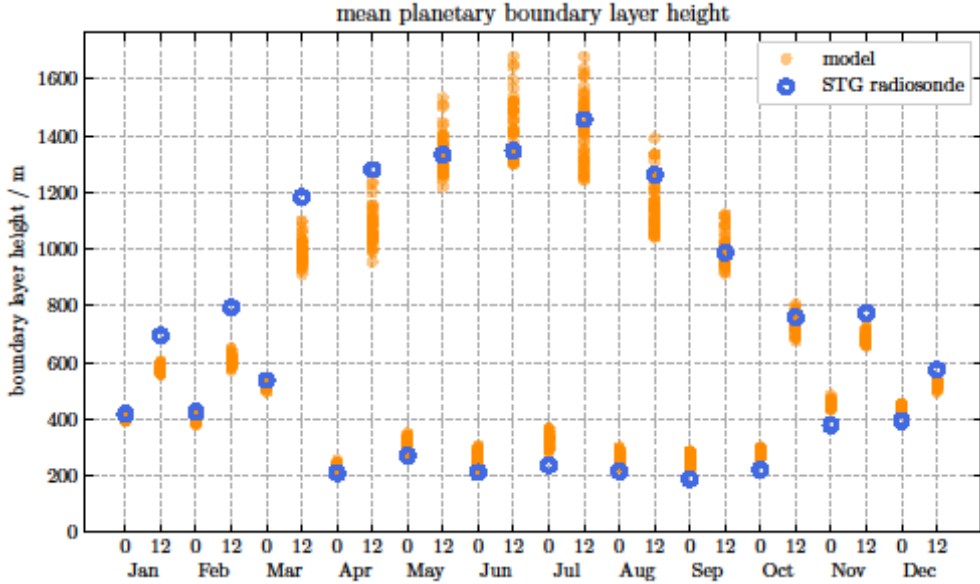

**Figure 12: Monthly mean boundary layer height at 0 h and 12 h local time for different land covers diagnosed from radiosonde observations at Stuttgart STG and from atmospheric profiles above grid boxes of CLM.**

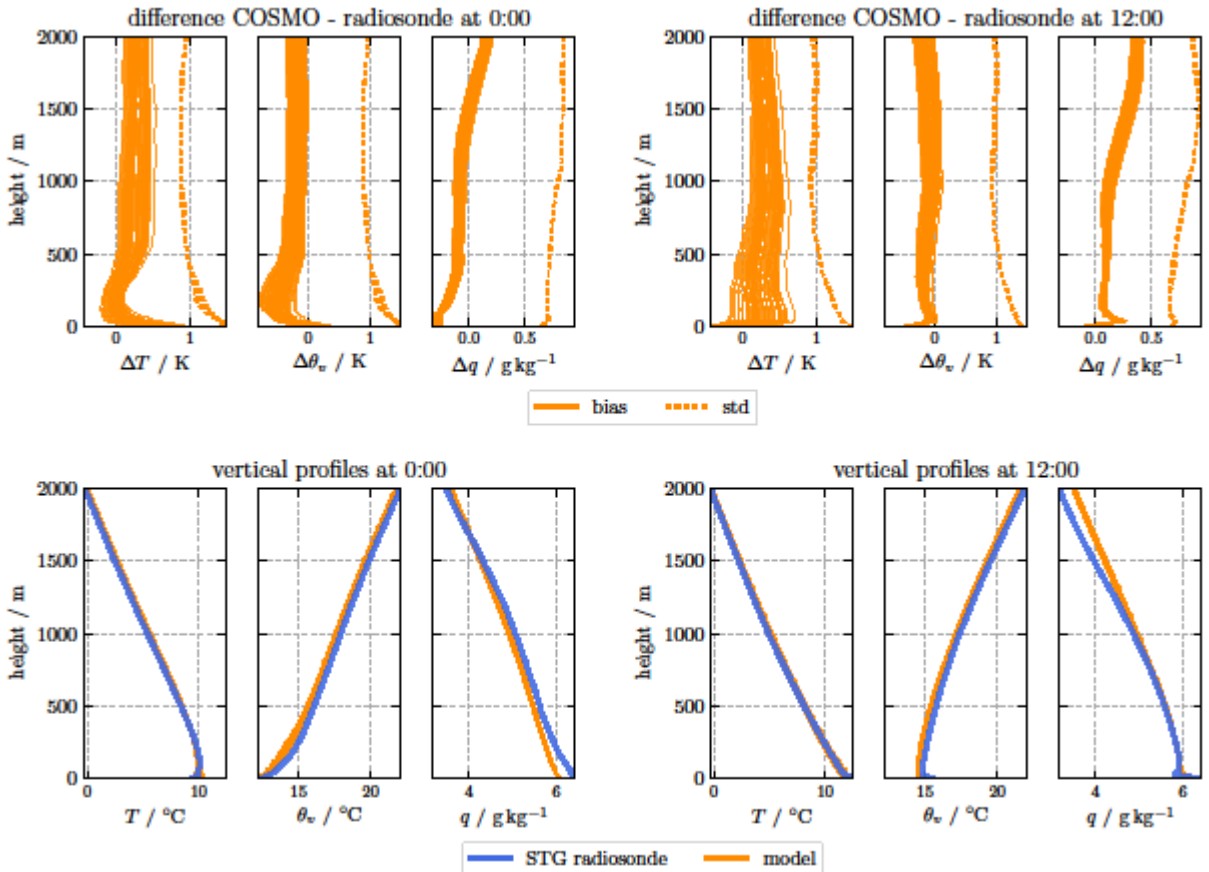

**Figure 13: Mean vertical profiles of temperature, virtual potential temperature, and specific humidity (top), and mean differences between modelled and observed data including the standard deviation of the differences (bottom). The experimental data are from the radiosonde data at STG and the simulated data from the grid boxes of the virtual catchment with different land cover (left: 0 h local time, right: 12 h local time).**

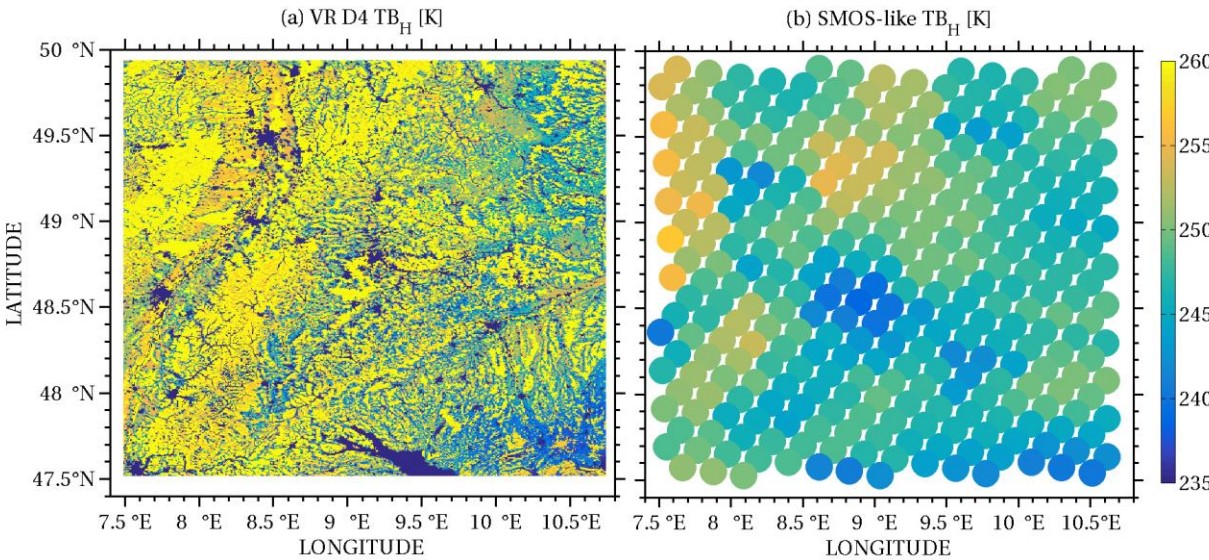

**Figure 14: Brightness temperature calculated by the application of CMEM (H-polarization) on the virtual-reality output on July 2nd 2011 (left) and its aggregation on the spatial resolution of the L1C data-product SMOS passive microwave radiometer (right).**

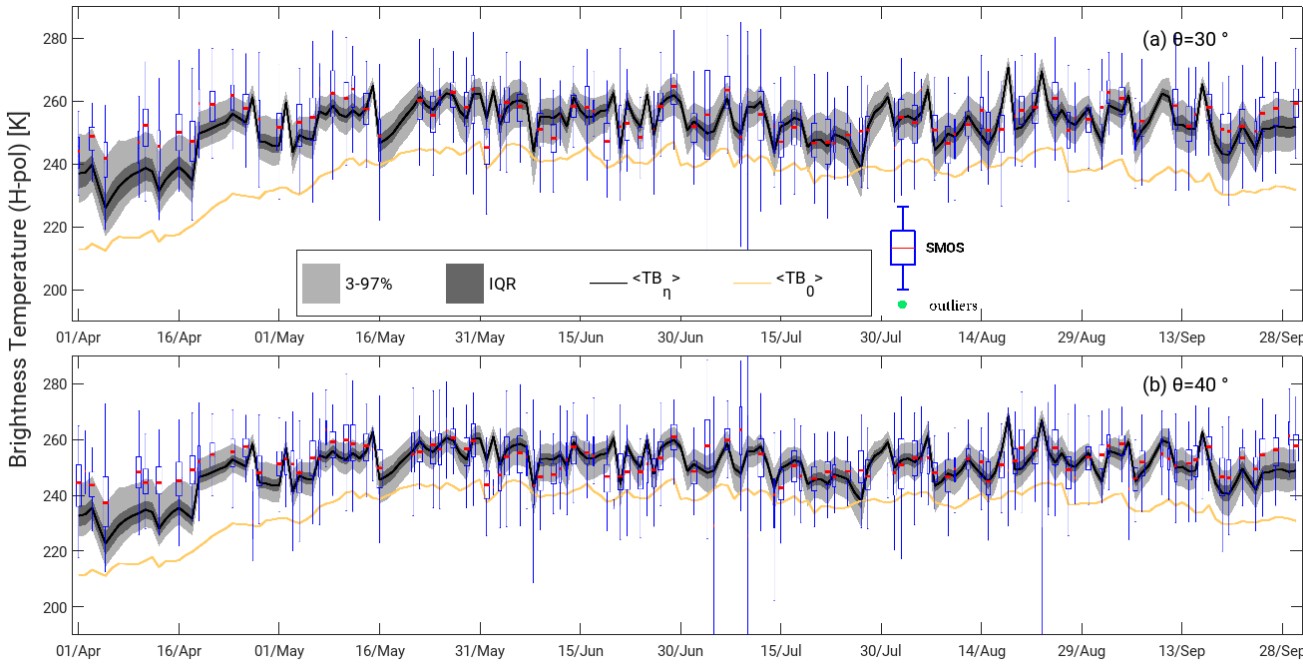

**Figure 15: Area-averaged L-band brightness temperature the period from April to September 2011 for an incidence angle of 30° (top) and 40° (bottom). The boxplots indicate the real SMOS observations averaged over the same domain. The black line is the median of the virtual observations simulated with CMEM. The dark-gray area corresponds to the inter-quartile range (IQR) while the light-gray area encompasses the 3 to 97% range. The orange continuous line indicates the brightness temperature without taking into account an assumed bias in surface soil moisture content (see text).**



**Figure 16: Hourly values river discharge at the gauging stations Rockenau (P1), Lauffen (P2) and Plochingen (P3) for the year 2007. Blue: observed; red: virtual catchment.**



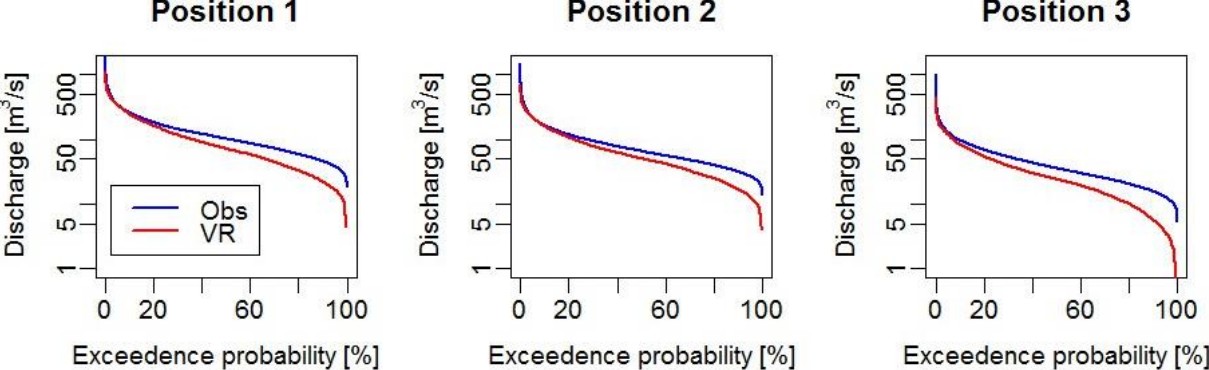

**Figure 17: Flow duration curve for the three stations for the three year time period based on. Blue: observations; red: virtual catchment.**

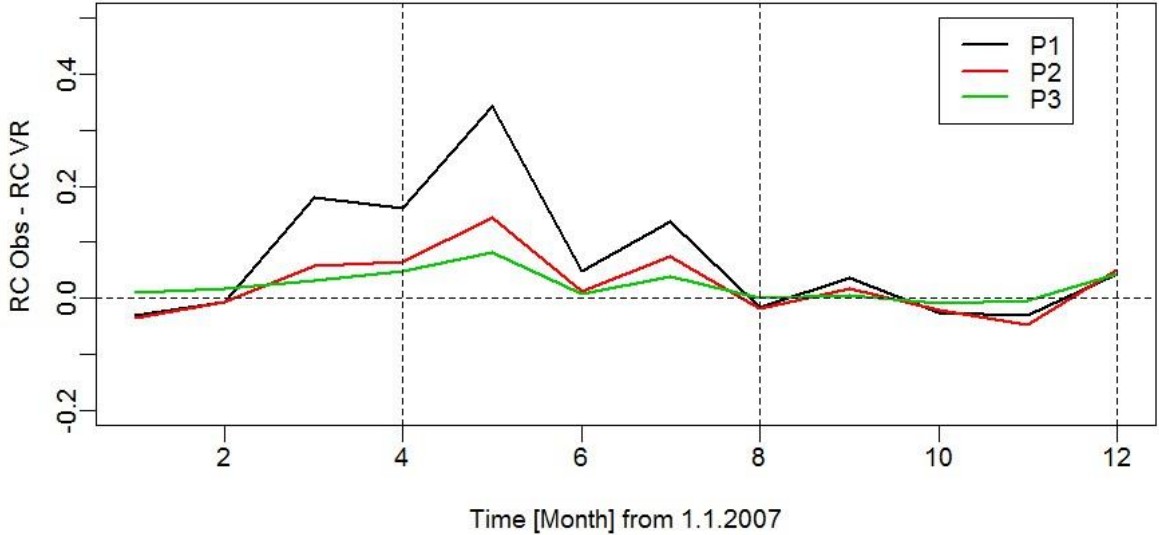

**Figure 18: Differences between the run off coefficient calculated for the three stations for the year 2007 based on observations and virtual catchment.**

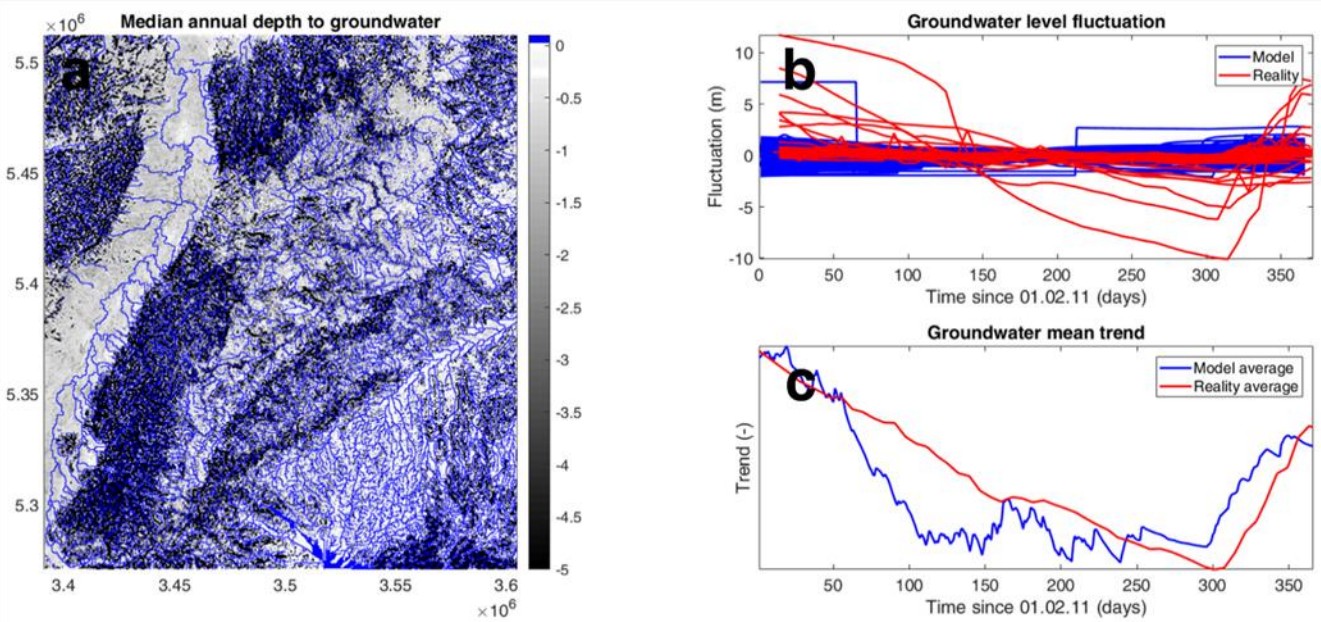

808

**Figure 19:(a) Mean groundwater table depth of the entire domain for the year ranging from 01.02.11 to 01.02.12, (b) groundwater fluctuations around a zero mean and (c) the total mean of all model cells and all real data points superimposed on top of each other to show the annual average trend. Please note that for readability of the figure, subfigure (a) is limited to a maximum depth of -5 m, while the underlying data ranged down to -88 m**







8000



**Appendix**
**7.1    Appendix Tables**
**Table A1: Values of porosity and hydraulic conductivity of rocks found in Baden-Wuerttemberg**

| Nr. | rock type | Ksat [m/h] | porosity fraction |
|---|---|---|---|
| 1 | Quarternary | 0.00100 | 0.3 |
| 2 | Tertiary | 0.00100 | 0.3 |
| 3 | Upper Jura | 0.00720 | 0.3 |
| 4 | Middle Jura | $10^{-7}$ | 0.3 |
| 5 | Lower Jura | $10^{-7}$ | 0.3 |
| 6 | Upper Triassic (Keuper) | 0.00036 | 0.3 |
| 7 | Middle Triassic (Muschelkalk) | 0.00180 | 0.3 |
| 8 | Lower Triassic (Buntsandstein) | 0.02160 | 0.4 |
| 9 | Upper Permian (Rotliegendes) | 0.00360 | 0.3 |
| 10 | New Red Conglomerate | 0.00100 | 0.3 |
| 11 | Bedrock/Granite | $10^{-7}$ | 0.3 |








**Table A2: Observed atmospheric variables at KIT and STG. Local time at STG is UTC+01.**

| dataset | quantity | temporal resolution | height above ground | data coverage |
|---------|----------|---------------------|---------------------|---------------|
| KIT | temperature | 10 min averages (resampled to 15 min) | 10 m, 100 m | 01/2007 – 12/2013 |
| | Incoming and outgoing shortwave radiation | | - | |
| | Incoming and outgoing longwave radiation | | - | 06/2011 – 12/2013 |
| STG | temperature | 12 h (11:45 h and 23:45 h local time) | vertical profiles (interpolated to model levels) | 01/2007 – 12/2013 |
| | dew point temperature | | | |
| | pressure | | | |
| | incoming shortwave radiation | 1 h averages | - | |
| | incoming longwave radiation | | - | |



