# Peer review of "Presentation and discussion of the high resolution atmosphere-land"

_Earth System Science Data, 2020_

## Referee Comment (RC1) · Anonymous Referee #1 · 7 Aug 2020

Review of Schalge et al. "Presentation and discussion of the high resolution atmosphere-land surface-subsurface simulation dataset of the virtual Neckar catchment for the period 2007-2015"

This study introduces a high-resolution dataset of the hydro-climatology of the subsurface, surface and atmosphere obtained with the coupled TerrSysMP model. It covers the area around the Neckar catchment in south-western Germany, and the time period 2007-2015. The dataset is validated in terms of several variables covered by respective observations, and is described to be a testbed for exploring land-atmosphere

interactions at high spatial resolution.

—————————-

Recommendation:

I think the paper requires major revisions.

I like the idea of introducing a comprehensively validated, modeled high-resolution land-atmosphere dataset, but regarding the present version of the manuscript and dataset I have several concerns which need to be addressed.

—————————

General comments:

(1) The applicability and purpose of the dataset is not entirely clear to me. The authors only briefly comment on this in the conclusions section, stating it could be used for exploring land-atmosphere feedbacks, investigating potential model simplifications, and data assimilation testing. Honestly, I do not see how the dataset can be useful in such analyses as (i) land-atmosphere interactions are known to be hardly robustly captured by models in general, and (ii) the testing of model simplifications and data assimilation would require different versions of the dataset with respective different model configurations in my opinion. In this context, I ask the authors to expand and clarify their discussion on the applicability of this dataset.

(2) CLM3.5 is used here as land surface model. This is outdated. By now, CLM5 is clearly more advanced in terms of simulating processes related to vegetation and hydrological dynamics at the land surface (Lawrence et al. 2019).

(3) Throughout the manuscript there are various inaccurate statements limiting the reproducibility of the model simulation (e.g. 'increased by about', 'increased by approximately', 'set considerably higher', 'needed to be increased from its standard values', various pedotransfer functions used without explaining criteria, see also respective

**ESSDD**
[Figure]

specific comments below). More information needs to be provided in each of these cases to ensure the reproducibility of the entire analysis, either in the manuscript or in an appendix.

(4) There are several arbitrary choices made throughout the study which need to be (better) motivated. This includes modifications of the modeling setup of which the purpose is not clear or the magnitude is arbitrary (e.g. 20% increase of sand fraction, ignoring of karst layers, conceptualization of alluvial layers as gravel and bedrock layers including the assumption(s) of values for various involved parameters, modifications of the LAI data). When making these modifications to adapt the model behavior in particular respects (more sandy soils to enhance infiltration) it should be kept in mind that even if the particular purpose is fulfilled, the land-atmosphere system is highly interconnected such that unforeseen side effects can occur. Further, the arbitrary choices include the approach(es) used to validate the modeled dataset (e.g. spatial averaging of model data across 25 grid cells for validation of atmospheric boundary layer characteristics, seemingly random time intervals of the soil moisture, evapotranspiration and runoff validations).

(5) Soil types are an important ingredient for hydro-climatological model simulations. The downscaling-based derivation of soil types in this study is (i) difficult to understand and (ii) contains several assumptions which are not motivated, among which are the amount of considered 1995 locations, the 20% increase of sand fraction (see above), the choice of an exponential model, the choice of conditional co-simulation versus kriging, the focus on first three soil horizons (first means uppermost I guess?). I wonder what is the impact of the choices made here on the final dataset?

(6) The validation of the model simulation in terms of evapotranspiration is very limited. While it is reassuring that the ET and groundwater dynamics are broadly coupled according to expectations this is not a quantitative assessment. The modeled ET could instead (or additionally) be compared with state-of-the-art evapotranspiration datasets such as GLEAM (Martens et al. 2017) or FLUXCOM (Jung et al. 2019) at larger spatial

scales.

(7) I like the comprehensive validation of the dataset in terms of several variables - an overview table summarizing the determined strengths and weaknesses would be helpful for users I think.

(8) There are too many figures in my opinion, diluting the main messages. Figures 3-5, 7, 14, 17 could be moved to supplementary, and Figures 9 & 10 could be combined.
* * *
Specific comments:

line 35 and throughout: 'simulated' would be more straightforward than 'virtual', using such terminology the term 'real' (line 34 and throughout), referring to observations, can be removed from the manuscript

line 56: test a disaggregation method

line 75-78: you do not aim to reproduce to observed catchment dynamics but still validate the model in some respects - this seems contradictory to me; what is the aim here if not validating the model against observations? how useful is a modeled dataset for the community if is not resembling observations?

lines 139 & 145: the chosen time period and simulation catchment/area are not motivated

line 172: please give more information on the 'software restriction'

line 183: please give more information on the location of the grid cells and the artificial elevation modification to ensure reproducibility, here or in an appendix

lines 195-196: 'about 20%' & 'about 3.3', please be more accurate

lines 194-197: in the abstract of the Tian et al. 2004 paper I found "On average, the model [...] overestimates FPAR over most areas in the Northern Hemisphere compared to MODIS observations during all seasons except northern middle latitude summer." "The MODIS LAI is generally consistent with the model during the snow‐free periods..." which makes me wonder why the authors modify LAI in summer? Further this could create jumps in the LAI time series from May-June and August-September. More importantly, you state here that LAI is used "for the year 2008". Does this mean there is no interannual vegetation variability? This would affect evapotranspiration and thereby many related variables and would need to be stated as a serious shortcoming.

line 204: please give the spatial resolution instead of the scale 1:1000000

line 221: 'approximately 20%', please be more accurate; further, and more importantly, please motivate this modification and its magnitude

lines 229-231: please give detailed information on where which pedotransfer functions have been used

lines 239 & 242: repetition of the information that karst is not considered

lines 239-240: 'to avoid the manifold hydrological challenges related to its modeling', please be more specific here, also please comment on the impact of this simplification of the approach on the final dataset

line 245: if these alluvial bodies are so relevant, why does this study use datasets which do not include them?

line 249: evapotranspiration errors in models can be significant and might be underestimated here

section 4: please discuss for the performed validation analyses how the determined performance of the study dataset compares generally with the performance of other regional climate models im similar hydro-climatic regions

line 277: 'simulated realistically', please give more details here on how this is quantified

line 301: 1km2, is this referring to the spatial resolution being 1km x 1km?

line 323: 'quite well', please be more specific and objective. Further, in Figure 9a between 6h-17h the pattern in the model is actually opposite to that of the observations, I would not refer to this as fitting "quite well".

line 331: the potential (dis)agreement of simulated and actual land cover can be checked using high-resolution land cover datasets such as provided by ESA CCI

line 343: how are the temperature standard deviations determined?

line 361: 'very well', please be more specific and objective

line 367: why not using the ESA CCI soil moisture dataset derived from observations of various satellites for this validation?

lines 394-395: I do not really understand why this daily matching is applied here? Also it is not clear how this is done.

lines 391 & 396: I guess you are referring to Figure 15 here, not Figure 16 as stated.

line 415: 'adequate agreement', please be more specific and objective

line 422: 'will always be replaced' needs to be toned down in my opinion

line 415: 'good distribution', please be more specific and objective

lines 447-450: I do not understand how the "fluctuations" are "scaled". Do you divide by the inter-annual standard deviation to obtain normalized anomalies (or z-scores)? If so, please name it this way as the term "fluctuations" is rather unclear.

line 450: I guess this should be "according to Figure 19b" and not 19c?

lines 451-452: How is this trend computed?

line 454 and following: I like this discussion of limitations and issues

line 458: to me it seems three challenges being discussed here (?)

lines 502-512: As there are multiple concrete ideas to improve the model setup and

consequently the dataset, why not implementing them before publishing this dataset? comment if this will be game-changers

Figure 2: This figure is the same as in the Gasper et al. 2014 paper, with the reference given in the caption. I think it is uncommon to use figures from previous papers, so I would remove this and only refer to the figure in the reference paper in the main text.

Figure 3: Maybe I missed that but what is domain 1?

Figure 4: "e+00" can be removed

Figures 4-6: Please label the color bars.

Figure 6: Please harmonize "evaporation" and "evapotranspiration" in the caption and the axis label. The same applies for the main text in section 4.1.

Figure 8: Values are quite far apart from color bars. Also, it would be nice to also express the difference as percentage.

Figure 11: y-axis label missing Please explain what is meant with the "temperature standard deviations"

Figure 13: Please specify from which times the reference radiosonde observations are taken. Further, please explain how the standard deviation is derived.

Figure 15: It would be insightful to quantify the agreement of the temporal dynamics with e.g. a correlation.

Figures 16-18: Please use the station names throughout instead of the position numbers.

Figure 19: Panel a is not labelled, as well as color bar and axes therein The terms "model" and "reality" are not consistent with the terminology used throughout the manuscript.

References:

Martens, B., Miralles, D. G., Lievens, H., van der Schalie, R., de Jeu, R. A. M., Fernandez-Prieto, D., Beck, H. E., Dorigo, W. A., and Verhoest, N. E. C.: GLEAM v3: satellite-based land evaporation and root-zone soil moisture, Geosci. Model Dev., 10, 1903–1925, doi:10.5194/gmd-10-1903-2017, 2017.

Jung, M., Koirala, S., Weber, U., Ichii, K., Gans, F., Camps-Valls, G., Papale, D., Schwalm, C., Tramontana, G., and Reichstein, M.: The FLUXCOM ensemble of global land-atmosphere energy fluxes, Scientific Data, 6, 74, doi:10.1038/s41597-019-0076-8, 2019.

Tian, Y., Dickinson, R. E., Zhou, L., Zeng, X., Dai, Y., Myneni, R. B., Knyazikhin, Y., Zhang, X., Friedl, M., Yu, I., Wu, W., and Shaikh, M.: Comparison of seasonal and spatial variations of leaf area index and fraction of absorbed photosynthetically active radiation from Moderate Resolution Imaging Spectroradiometer (MODIS) and Common Land Model, J. Geophys. Res., 109, D01103, doi:10.1029/2003JD003777, 2004.

Gasper, F., Goergen, K., Shrestha, P., Sulis, M., Rihani, J., Geimer, M., and Kollet, S.: Implementation and scaling of the fully coupled Terrestrial Systems Modeling Platform (TerrSysMP) in a massively parallel supercomputing environment-a case study on JUQUEEN (IBM Blue Gene/Q), Geoscientific model development discussions, 7, 3545-3573, doi:10.5194/gmdd-7-3545-2014,2014.

Lawrence, D. M., Fisher, R. A., Koven,C. D., Oleson, K. W., Swenson, S. C.,Bonan, G., et al. (2019). The Community Land Model version 5: Description of new features, benchmarking, and impact of forcing uncertainty, Journal of Advances in Modeling Earth Systems, 11,4245–4287, doi:10.1029/2018MS001583, 2019.

---

## Author Comment (AC1) · 29 Oct 2020

Dear Reviewer,

thank you for your comments and for the detailed suggestions. Below we provide our responses and how the manuscript will be modified accordingly.

> ***Recommendation:***
> *I think the paper requires major revisions. I like the idea of introducing a comprehensively validated, modeled high-resolution land-atmosphere dataset, but regarding the present version of the manuscript and dataset I have several concerns which need to be addressed.*

Thank you for your positive feedback regarding the overall purpose of the study and of the dataset. We regret to know that some parts of the manuscript were not clear, but we believe that comments and suggestions were very useful to address these issues and strengthen the discussion. Specific responses and how the manuscript will be improved are listed below.

> ***General remarks:***
> *1.) "The applicability and purpose of the dataset is not entirely clear to me. The authors only briefly comment on this in the conclusions section, stating it could be used for exploring land-atmosphere feedbacks, investigating potential model simplifications, and data assimilation testing. Honestly, I do not see how the dataset can be useful in such analyses as (i) land-atmosphere interactions are known to be hardly robustly captured by models in general, and (ii) the testing of model simplifications and data assimilation would require different versions of the dataset with respective different model configurations in my opinion. In this context, I ask the authors to expand and clarify their discussion on the applicability of this dataset."*

Indeed, we agree with the reviewer that the purpose of this dataset is a key characteristic that should be clear from the manuscript. Within our project, we are working on developing and testing data assimilation approaches. More specifically, we aim to disentangle the value of different observation types in fully coupled models. We used the fully coupled model to develop a virtual reality. We faced two important issues. First, the reliability of the created virtual reality. Second, the huge computational effort needed to run a simulation with the desired spatial and temporal resolution. We decided to work therefore on this study to 1) provide a comprehensive assessment of the reliability of the states and fluxes simulated by the fully coupled models, and 2) make not only model code, forcings and parameters available but also the complete model output to guarantee the reproducibility of the entire model. This should allow the use of this model setup by other groups to do targeted tests, evaluating for example the impact of a different spatial model resolution, or a specific simplification (for example, neglecting lateral flows in the subsurface). This is challenging because we aimed at mimicking the real world without doing a formal model calibration. We believe that the comparison with real observations within the area support the capability of the fully coupled model to reproduce the water and energy cycles and should provide the confidence to use the dataset.

The dataset will be used for data assimilation experiment in our project, but is available for anybody interested in data assimilation and modelling experiments. We will extract (virtual) observations from the virtual reality and perturb the observations with a measurement error. The effectiveness of any data assimilation algorithm can be quantified making it a powerful tool for the development of such algorithms. In addition, thanks to the availability of a fully coupled system, we will be able to check the

effect of assimilating observations in the different terrestrial compartments (e.g., assimilating groundwater level data and check effect on land-atmosphere fluxes like evapotranspiration).

In addition to this primary purpose, this dataset can be exploited for model simplification, to test reconstruction methods or define monitoring networks. Examples from our work are listed:

Baroni, Gabriele, Bernd Schalge, Oldrich Rakovec, Rohini Kumar, Lennart Schüler, Luis Samaniego, Clemens Simmer, and Sabine Attinger. "A Comprehensive Distributed Hydrological Modelling Inter-Comparison to Support Processes Representation and Data Collection Strategies." *Water Resources Research*, January 17, 2019. https://doi.org/10.1029/2018WR023941.

Lv, Shaoning, Bernd Schalge, Pablo Saavedra Garfias, and Clemens Simmer. "Required Sampling Density of Ground-Based Soil Moisture and Brightness Temperature Observations for Calibration and Validation of L-Band Satellite Observations Based on a Virtual Reality." *Hydrology and Earth System Sciences* 24, no. 4 (April 17, 2020): 1957–73. https://doi.org/10.5194/hess-24-1957-2020.

Haese, B., S. Hörning, C. Chwala, A. Bárdossy, B. Schalge, and H. Kunstmann. "Stochastic Reconstruction and Interpolation of Precipitation Fields Using Combined Information of Commercial Microwave Links and Rain Gauges." *Water Resources Research* 53, no. 12 (2017): 10740–56. https://doi.org/10.1002/2017WR021015.

In Baroni et al. (2019) a similar dataset created with TerrSysMP is compared to a distributed, conceptual-based hydrological model. The comparison was able to identify where and when simulations yielded similar results (and as such the simplified model could be used instead of the complex one) and where differences emerged (and as such identifying the need of model improvements and measurements). In Lv et al. (2020) the virtual reality was used to identify optimal sampling strategies for capturing soil moisture dynamics at a desired accuracy and precision. This is a basis for designing cost-efficient monitoring schemes. Haese et al. (2017) introduced a method to reconstruct precipitation fields combining precipitation measured by rain gauges and commercial microwave links. The synthetic dataset allowed the comparison of the reconstructed fields and its estimated error against the virtual truth. These are just few examples of how the data set can be used to test different methods or modeling strategies.

In addition, it is important to point out that indeed land-atmosphere feedbacks are difficult to capture by models. The advantage of our coupled modelling approach is that we replace the hydrology of the land surface model CLM3.5 with an approach where lateral subsurface flows (soil and, especially important, groundwater) are considered as well, and also lateral flows along the land surface (streams). As such, it is a state of the art model for representing land-atmosphere interactions, especially for the water and energy cycles. We agree that using CLM3.5 instead of CLM5.0 implies that the representation of vegetation and biogeochemical cycles in land-atmosphere interaction it is not the best we can do at the moment. However, a coupled land-atmosphere model using CLM5.0 and at the same time a full subsurface hydrology including lateral flows is not available yet.

We will add all these clarifications as to how we want to use this dataset ourselves and how other people can use it in the new version of the manuscript. We will clarify this in the introduction of the manuscript.

*2.) "CLM3.5 is used here as land surface model. This is outdated. By now, CLM5 is clearly more advanced in terms of simulating processes related to vegetation and hydrological dynamics at the land surface (Lawrence et al. 2019)."*

We agree that CLM has seen several improvements in the last years. Accordingly, an updated version of TerrSysMP with CLM5 is also in development and being tested. However, it is important to note that the upgrade to CLM5 would have only small impact on the dataset presented and discussed in this study. The main reason is that the hydrology including overland flow and lateral water flows in the subsurface including the representation of groundwater are being replaced by ParFlow. So the simplicity of CLM3.5 in that regard is not an issue. In addition, other CLM5 improvements, such as C and N cycles are not the focus of our dataset. For these reasons, while we agree that the use of CLM3.5 could be seen as outdated, the actual dataset created with the coupled modelling system is not, given the mechanistic representation of the terrestrial water cycle. We will better clarify this in the revised version of the manuscript.

*3.) "Throughout the manuscript there are various inaccurate statements limiting the reproducibility of the model simulation (e.g. 'increased by about', 'increased by approximately', 'set considerably higher', 'needed to be increased from its standard values', various pedotransfer functions used without explaining criteria, see also respective specific comments below). More information needs to be provided in each of these cases to ensure the reproducibility of the entire analysis, either in the manuscript or in an appendix."*

We will improve the description with more details in the appropriate sections in a revised version of the paper. We want to point out that the forcing and namelist files are part of the dataset as supplementary files, so the exact setup of our models can be re-created easily. A better link to them also within the text will be provided in the new version of the manuscript.

*4.) "There are several arbitrary choices made throughout the study which need to be (better) motivated. This includes modifications of the modeling setup of which the purpose is not clear or the magnitude is arbitrary (e.g. 20% increase of sand fraction, ignoring of karst layers, conceptualization of alluvial layers as gravel and bedrock layers including the assumption(s) of values for various involved parameters, modifications of the LAI data). When making these modifications to adapt the model behavior in particular respects (more sandy soils to enhance infiltration) it should be kept in mind that even if the particular purpose is fulfilled, the land-atmosphere system is highly interconnected such that unforeseen side effects can occur. Further, the arbitrary choices include the approach(es) used to validate the modeled dataset (e.g. spatial averaging of model data across 25 grid cells for validation of atmospheric boundary layer characteristics, seemingly random time intervals of the soil moisture, evapotranspiration and runoff validations)."*

Our aim was to generate a virtual reality for the atmosphere-land surface-subsurface system which is close to the reality. Therefore, a detailed comparison with measurements was made. It is usual in land surface and subsurface modelling that systematic deviations occur between model simulations and measurement data, which are related to specific parameter settings. It is therefore not surprising that we also find these systematic deviations, and in order to reduce these systematic deviations we made some additional adjustments:

- 20% increase of sand fraction: preliminary simulations with the model showed a wet bias related with too high soil moisture and groundwater levels by using the soil parametrizations derived by the original soil map. This wet bias has been detected in many other studies using Parflow or similar hydrological models (Shrestha et al., 2015, 2018) and it is generally attributed to the need of effective parametrizations that account for the spatial resolution of the model (see also discussion in the manuscript on that topic). Usually, few specific hydraulic parameters like the hydraulic conductivity or the Van Genuchten n value are modified to account for that. However, we found that these changes would create some inconsistencies in the modelling settings. This is specifically true in our coupled modelling framework due to a strong dependency between thermal and soil hydraulic properties. For this reason, instead of increasing a single specific parameter like e.g., hydraulic conductivity to account for effective model resolution, we decided to change the texture percentages. Accordingly, by applying pedotransfer functions to estimate hydraulic and thermal parameters, all the soil parameters are modified consistently. In the specific, the increase of the sand percentage of 20% has been applied to increase the soil infiltration capacity but to minimize the change in the soil classes detected in the original soil map.
- Elimination of karst. Karst cannot be represented well in standard hydrogeological models. We prefer not to claim that we can model karstic areas well with this approach.
- Conceptualization of alluvial layers as gravel and bedrock layers. Even though the Neckar river valley is (with some exceptions) quite narrow and has rather small alluvial layers, we found that alluvial layers needed to be introduced to increase subsurface water flow and reduce soil moisture contents in river valleys. This can be considered an additional calibration step to simulate groundwater levels and soil moisture contents closer to reality.
- Modification of LAI-data: this was done to remove bias from remotely sensed MODIS-data especially for winter time. This larger bias of MODIS-data is for example related to snow covered needleleaf forests.
- Spatial averaging of model data across 25 grid cells for validation of atmospheric boundary layer characteristics. This is done because we only considered the most dominant land-use type for a grid cell. However, in reality we would have a mix of several land use types and by choosing 25 grid cells we can consider this in our analysis.
- Time intervals for soil moisture, evapotranspiration and runoff validations. For each of the mentioned variables we want to show the most important aspect of the modelled behavior and we selected the periods accordingly. For river discharge showing we selected three years with relevant differences in the weather conditions (wet and dry periods). Some specific results are shown for one year as example to visualize the river discharge under changing seasonal conditions. Please also note that there is an error in the text where the time period mentioned is not the same as the one shown in the figure, this will be corrected. As for ET we show a basic analysis of the relation between ET and groundwater level and then show a selection of representative gridpoints to explain the observed behavior. ET and near-surface soil moisture react very fast to rainfall events and thus a smaller time interval is needed to analyze this. In addition, the chosen soil moisture time interval is related to the frequency of satellite overpasses.

These clarifications will be made in the different sections of the new version of the manuscript.

*5.) "Soil types are an important ingredient for hydro-climatological model simulations. The downscaling-based derivation of soil types in this study is (i) difficult to understand and (ii) contains several assumptions which are not motivated, among which are the amount of considered 1995 locations, the 20% increase of sand fraction (see above), the choice of an exponential model, the choice of conditional co-simulation versus kriging, the focus on first three soil horizons (first means uppermost I guess?). I wonder what is the impact of the choices made here on the final dataset?"*

We thank the reviewer to give us the opportunity to provide more details regarding the generation of the soil types. Indeed, the method to downscale the original soil map underwent several tests. Some of them have been described in (Baroni et al., 2017). We have referred to that paper also within the manuscript, but we summarize below the main steps for the sake of clarity.

The original soil map with necessary data for running the model is quite coarse with the areas of most of the soil polygons above 20 km2 (Figure 1).

[Figure]

Figure 1: original soil map of the area. Please note that these figures do not represent the entire domain of the VR1 but they are reported only to show the main characteristics.

Based on that, different methods have been tested and compared to the new so-called conditional point method (CPM). The results are depicted in Figure 2.

[Figure]

Figure 2: soil clay map created with different methods. From left: random error (RE) method where nominal value of each soil type has been perturbed, spatially correlated (SC) method where a spatial random variable has been superimposed to the original soil map (black line) and, conditional point (CP) method where conditional points are extracted and used in the conditional simulation.

The results showed that the CP method introduces uncertainty only at small spatial scales while the longer spatial patterns are preserved without sharp changes between the soil units. For this reason, this method has also been selected within the present study. Please also note that we did not consider the kriging approach as kriging removes small-scale variability which was considered an important feature to be considered in the dataset and coupled modelling.

The conditional point CP method is described in the next Figure 3.

[Figure]

Figure 3: sketch of the method to downscale the soil map (a transect is depicted as example)

Starting from the original soil map, (step C1) soil samples are selected, (step C2) a variogram model is fitted and (step C3) a conditional simulation is created. In total 1995 locations have been used to mimic

a realistic number of soil samples that can be collected for creating a soil map. More specifically a density of 1 sample per 5 km$^2$ is used. Some tests conducted for different sampling densities did not change the results significantly. The procedure will be discussed in greater detail in the new version of the manuscript. In contrast, the selection of the variogram model did not underwent a detailed analysis but was selected to provide a stronger short scale variability in contrast to the Gaussian variogram. This will be also acknowledged in the new version of the manuscript. Finally, the use of the first upper soil horizons has been considered because they represent the main soil horizons of the root zone (horizons A, B and C). Additionally, the use of only the first three horizons allowed us to have a variable soil depth under which we imposed the rocks. The original soil map provides more soil layers which would have resulted in a uniform soil depth of 2m.

As previously discussed, all these features have been selected after several tests conducted with the fully coupled model or with simpler configurations (not coupled, short period, limited areas), finally resulting in more realistic dynamics. We did not completely evaluate the impact of all these decisions on this dataset. This would require a detailed sensitivity analysis that is beyond the scope of the present study and also not feasible from the computational point of view. Still, we thank the reviewer for the comments and will clarify these aspects of the simulations in the new version of the manuscript.

> *6.) "The validation of the model simulation in terms of evapotranspiration is very limited. While it is reassuring that the ET and groundwater dynamics are broadly coupled according to expectations this is not a quantitative assessment. The modeled ET could instead (or additionally) be compared with state-of-the-art evapotranspiration datasets such as GLEAM (Martens et al. 2017) or FLUXCOM (Jung et al. 2019) at larger spatial scales."*

We have limited the scope of the ET analysis since a general analysis of how TerrSysMP performs with respect to ET was already done (Shrestha et al., 2017 https://doi.org/10.1016/j.jhydrol.2018.01.024). Since our land surface is different from the real land surface (the land-use is similar, but restricted to a dominant land use type per gridcell and LAI is fixed without interannual variability) a comparison to any real world dataset would be biased and it would be very hard to figure out if the differences seen in the analysis are due to the difference in land-use, neglecting inter-annual variability or differences related to the model biophysics. In addition, the datasets GLEAM and FLUXCOM are not measured values, but also affected by strong model assumptions (GLEAM) and/or interpretation (e.g., energy balance errors at EC-sites) and interpolation (for FLUXCOM) of data.

> *7.) "I like the comprehensive validation of the dataset in terms of several variables - an overview table summarizing the determined strengths and weaknesses would be helpful for users I think."*

Thanks for the suggestion. We will add such a table to the manuscript.

> *8.) "There are too many figures in my opinion, diluting the main messages. Figures 3-5, 7, 14, 17 could be moved to supplementary, and Figures 9 & 10 could be combined."*

We will reduce the number of figures and move to supplementary. Please note that there are also many specific comments for the figures, automatically leading to a reduction in the number of figures as suggested.

**Specific comments**:

> "line 75-78: you do not aim to reproduce to observed catchment dynamics but still validate the model in some respects - this seems contradictory to me; what is the aim here if not validating the model against observations? how useful is a modeled dataset for the community if is not resembling observations?"

The aim of the validation was to show that the model system, despite its simplifications, is still able to simulate all the important core processes adequately. Therefore, we compared many of our output variables to observations to see if we had large biases or other undesirable behavior by the choices of our setup. However, a full reproduction of all measurement data is not possible as this requires the calibration of an integrated atmosphere-land surface-subsurface model, which still has not been documented in the literature. In summary, we want to check that systematic biases are small, but are concerned with random deviations between model and measurements.

> "lines 139 & 145: the chosen time period and simulation catchment/area are not motivated"

We have chosen the Neckar because it has varying topography and land-use and typical central-European climate. A further motivation to choose the Neckar catchment was that groundwater levels are relatively high so that groundwater-atmosphere feedbacks are more likely and prominent in this catchment. We wanted to have a long simulation time period while still having boundary data from a convection-permitting model since our domain is rather small and a huge boundary zone would have been needed if forcing from a coarser model was used. 2007 was thus the first year we had access to such a forcing and was chosen as the start of the simulation for that reason alone. In the revised version of the paper we will clarify these points better.

> "line 172: please give more information on the 'software restriction'"

This restriction was actually a combination of software and hardware. The machine we used was optimized for using many relatively slow cores. As such the only way to run the simulation was to use many cores at a time. As the number of grid cells increases so does the number of cores and at a certain point the model would no longer run.  We never found the root cause but the result was a limit to the domain size in terms of grid cells. Disabling I/O did increase the limit before the model would stop running but doing so is obviously not feasible. We strongly believe that this issue was related to the specific setup of the computer we used and is likely not to occur on more modern machines with higher per-core performance.

> "lines 194-197: in the abstract of the Tian et al. 2004 paper I found "On average, the model [...] overestimates FPAR over most areas in the Northern Hemisphere com- pared to MODIS observations during all seasons except northern middle latitude summer." "The MODIS LAI is generally consistent with the model during the snow-free periods..." which makes me wonder why the authors modify LAI in summer? Further this could create jumps in the LAI time series from May-June and August-September. More importantly, you state here that LAI is used "for the year 2008". Does this mean there is no interannual vegetation variability? This would affect evapotranspiration and thereby many related variables and would need to be stated as a serious shortcoming."

Since we do not use MODIS land-use but CORINE, we had to re-map the MODIS observations to the CORINE land-use map. As such you would have a range of values for each land use type. We wanted to

keep things simple and used just one value for each land use type at a given timestep. So, for the summer we chose values from the upper end of that range while only larger changes were made to LAI for needle leaf trees in winter. There are no jumps in the dataset. All years have the same LAI-cycle, so there is no interannual variability. We will acknowledge this in the revised version of the paper.

> "lines 239-240: 'to avoid the manifold hydrological challenges related to its modeling', please be more specific here, also please comment on the impact of this simplification of the approach on the final dataset"

The most obvious impact is the one of subsurface cave systems. These can be vast while being subgrid scale at the same time. They may only have an impact locally, or be connected to a larger system of caves that can re-distribute groundwater going as far as re-distributing smaller rivers (the Danube sinkhole is one example). They also can go deeper underground than what we are modelling. However, some regions may not be affected by the underground caves if they are not connected to the surface at that point while other regions are greatly impacted. Detailed maps of the cave system are not available. Even if such maps would be available, it would be an enormous effort to implement them or find parameterizations which could replace the karst system. Finally, the impact on the near-surface soil hydrology will only matter locally and for much smaller areas. We argue that including karst areas is beyond the scope of this study given the huge amount of time needed to represent karst systems adequately in the subsurface. In addition, a deeper vertical extent of the model domain would be needed as well. We will provide a clearer formulation in the revised version of the paper.

> "line 245: if these alluvial bodies are so relevant, why does this study use datasets which do not include them?"

The alluvial bodies of the Neckar are very small since the Neckar is most of time confined to a rather narrow valley. With our rather coarse resolution we are barely able to capture these valleys accurately. The soil map, which we used as basis for our soil hydraulic properties, did not contain these alluvial bodies, as these bodies are not soil. On the other hand, the geological map is much coarser and does not include these fine structures, although they are from the subsurface flow perspective important. We first tried if we could disregard the alluvial bodies altogether, similarly to the karst features, but found out that this has a large impact even for rather small streams which made it necessary to include them and due to the small size of the valleys our chosen extent of three grid cells is already generous in most locations. We will make in the manuscript some additional clarifications regarding the alluvial bodies.

> "line 249: evapotranspiration errors in models can be significant and might be underestimated here"

Previous studies (Shrestha et al, 2017, https://doi.org/10.1016/j.jhydrol.2018.01.024 ) have shown that for a climate as ours transpiration is energy limited. Therefore, any difference to real values would be due to LAI differences. The study also shows that we are likely overestimating ground evaporation due to the resolution effect discussed. That means that our value of 30% subsurface flow is actually a conservative estimate.

> "section 4: please discuss for the performed validation analyses how the determined performance of the study dataset compares generally with the performance of other regional climate models in similar hydro-climatic regions"

This is beyond the scope of this study. We provide all states and fluxes from the subsurface to the atmosphere and a major contribution is that these states and fluxes are also provided for the subsurface, including groundwater levels, groundwater flows, lateral water flows in soils, and river discharge. We wanted that this virtual reality is realistic and not far away from reality, for all those compartments of the terrestrial system. The aim was not a comparison with other RCM´s.

> "line 331: the potential (dis)agreement of simulated and actual land cover can be checked using high-resolution land cover datasets such as provided by ESA CCI

While we can check this, the mismatch will remain (and is mostly related to resolution effects and our choice to only use one land-use type per grid cell) and needs to be considered when comparing our results to actual observations. Attributing differences is much more difficult which is why we take the approach to average them out as much as possible.

> line 343: how are the temperature standard deviations determined?"

This standard deviation can be considered the mean absolute difference between the observations and our modelled results. This is done for each time of day separately.

> "line 367: why not using the ESA CCI soil moisture dataset derived from observations of various satellites for this validation?"

The remotely sensed soil moisture cannot really serve for validation. The simulated soil moisture content will be as reliable as the remotely sensed soil moisture content, so we prefer not to use ESA CCI for validation.

> "lines 394-395: I do not really understand why this daily matching is applied here? Also it is not clear how this is done."

The soil moisture overestimation by the virtual reality is largely responsible for the significant bias found when comparing synthetic with real observations of brightness temperature (Figure 15 orange-line). This soil moisture overestimation is not constant along the year, therefore a daily soil moisture correction factor was found in order to apply a correction factor to the soil moisture input for the radiative transfer simulation. As indicated in line 392, the correction factor is found by matching the cumulative distribution functions of soil moisture from virtual reality and satellite retrievals, this correction factor is then used to simulate the adjusted brightness temperature which is shown by black lines and shaded IQR area in Figure 15 a and b.

> "lines 447-450: I do not understand how the "fluctuations" are "scaled". Do you divide by the inter-annual standard deviation to obtain normalized anomalies (or z-scores)? If so, please name it this way as the term "fluctuations" is rather unclear."

The reviewer makes a fair point that this section was rather unclear. For Figure 19b, we simply compare the groundwater observations when the mean over the plotted period is removed, hence all lines in the plot have a zero mean. Apart from that, the figure is not scaled or shifted. For Figure 19b, scaling means that both the magnitude and the mean is manually shifted so that both curves can be compared in the same plot. Therefore, the y-axis on the figure lacks units and we simply talk about a trend. Hence, no formal scaling technique has been used. In a revised manuscript, this section will be better explained.

"lines 451-452: How is this trend computed?"

For the simulation this is the area-averaged changes of groundwater level over the full model domain, while for the observation wells it is the arithmetic average over all wells. As for the obvious difference between the two and the hence required scaling to compare them, please see the answer to the question above.

"lines 502-512: As there are multiple concrete ideas to improve the model setup and consequently the dataset, why not implementing them before publishing this dataset? comment if this will be game-changers"

Most of these changes are related to resolution. Going from 400m to 200m or 100m for the land surface and subsurface would obviously improve several aspects of the simulation, specifically for river discharge. However, such a high resolution is currently hardly feasible and computing resources would not be available to repeat the simulations at such a high resolution. We already went at the limits of what is currently feasible. Other improvements can be applied with less effort but still would require the entire simulation to be re-run, since they were developed after the simulation was done (as a way to improve the results we saw actually). While these changes would improve the simulation, we do not expect results to improve a lot. The only exception may be river discharge during flood events. Again, re-running this very large simulation (including a new spin-up) would consume a very large amount of compute resources. In the future with much improved computational power it would be of interest to do an updated version of such a run using ICON instead of COSMO, CLM5.0 instead of CLM3.5 and a GPU-based ParFlow version in order to see improvements for all terrestrial compartments. However, this new version of TerrSysMP (ICON-CLM5.0-ParFlow-GPU) is currently under construction and not ready in the next six months.

In addition to the specific comments above there are several more comments of minor deficiencies that can easily be fixed in an updated version of the manuscript. We did not include all of these here specifically.

---

## Referee Comment (RC2) · Anonymous Referee #2 · 20 Apr 2021

The paper titled "Presentation and discussion of the high resolution atmosphere-land-surface-subsurface simulation dataset of the virtual Neckar catchment for the period 2007-2015" introduces a comprehensive simulation of the land/atmosphere using the TerrSysMP dataset over the Neckar catchment between 2007-2015. My first impression of the paper was that indeed the authors have put in extensive work to create a comprehensive virtual reality of the Neckar catchment. The methods are comprehensive and the results are sufficiently developed. However, I still am left with the question

of what is this data going to be used for beyond the work done by the authors. From my understanding, a publication in ESSD should be a dataset that one could expect to be used extensively by the larger community. If this covered the entire country or Europe then that would be a different story, but I just don't see it as is. I still believe the work should be published but the paper would greatly benefit by putting it all in a greater context. I suggest the discussion provide a much more detailed overview of what this data could be used for and why it is necessary to have a unique dataset instead of just having each researcher rerun the simulations. In essence, it would be nice to know what this dataset could provide the larger scientific community 5-10 years from now. Unless that argument is compelling, it is hard to argue for the need for this dataset to be published.

---

## Author Comment (AC2)

Dear Reviewer2,

thank you very much for your review and comments. You seem to be mostly concerned with the usefulness of the dataset for the target audience:

> ***Original remark***:
> *1.) " However, I still am left with the question of what is this data going to be used for beyond the work done by the authors. From my understanding, a publication in ESSD should be a dataset that one could expect to be used extensively by the larger community. If this covered the entire country or Europe then that would be a different story, but I just don't see it as is. I still believe the work should be published but the paper would greatly benefit by putting it all in a greater context. I suggest the discussion provide a much more detailed overview of what this data could be used for and why it is necessary to have a unique dataset instead of just having each researcher rerun the simulations. In essence, it would be nice to know what this dataset could provide the larger scientific community 5-10 years from now."*

Your remark is very similar to the one of the first reviewer, to which we responded already when revising the manuscript; but obviously we still need to make the case stronger.

First, we want to address the concern of a too small domain used. Given the complexity of our model system driven by the challenge to determine the evolution of the water and energy fluxes in the land-atmosphere system as complete as possible with a most advanced model system able to most directly simulate all relevant processes, we are limited by the currently available computational resources. Our wish to create a dataset of several years in order to allow for climatological analyses, we were forced to restrict ourselves spatially. Given the new hard- and software developments such as e.g. GPU computing, such simulations will potentially be possible for the country and/or Europe probably in about a decade. However – even then – problems will arise on the availability of sufficiently resolved and homogeneous information on the soil and sub-soil required for such simulations.  We doubt that the quality of the respective data sets for the German state of Baden-Würtemberg, which we used in our work, will be available over Europe in a similar resolution and/or granularity. Under such conditions the results of spatially much more extended simulations would most probably signal more the different data sets than the processes – and thus restrict any analysis to the regions with consistent soil and – sub-soil data.

We see a range of applications for our data sets.  Users may repeat our simulations with their own model system or use our system and experiment with alternative or improved input data or parameterizations and evaluate according sensitivities by comparison with our simulations, which would serve as the benchmark. This is currently not possible, because similar almost decade long fully coupled simulations of comparable complexity are not available but will for sure in the future.  Since the information provided by our data set does allow users to apply observation operators for pseudo-observations, data assimilation experiments in the coupled system can be performed, which is what we are currently doing.

Since it was our goal to use the highest spatial grid resolutions and to reduce parameterization as much as possible under the computational constraints posed by currently available IT resources, the data set can be used to perform daily, seasonally and long-term analyses of intra- and intercompartmental water and heat energy fluxes and budgets in the land-vegetation-atmosphere system. The allows e.g. for

statistical analyses of the relations between the state and evolution of the soil moisture and temperature at any layer and the state of the atmospheric boundary layer and even precipitation patterns and vice versa taking into account also arbitrary lag-times. Given the diversity of landscape and land use contained in our simulations the dependency of such relations on the latter and on season can also be distilled from the data set. Since we cover a rather long time period and – given the comprehensiveness of the model system – large area, users can restrict such analyses also to sub-areas (e.g. smaller sub-catchments) or interesting time periods, certain times of the day, or seasons. Interesting examples are e.g. the dependency of convection in the atmosphere on the detailed state of the land and vegetation state and its heterogeneity in rather realistic settings or e.g. the magnitude of canopy evaporation after rain events at certain times of the day for a range of wind and temperature conditions.

If re-simulations with other model setups e.g. different parameterizations or spatial resolution are performed sensitivities of above relations to model configurations or parameterizations can be determined including e.g. failures in reproducing true or even threats in generating wrong connection can be detected, e.g. concerning the reaction of the atmosphere to certain features of the land. For our ongoing ensemble-based data assimilation experiments, we use due to prohibitive IT resources runs at lower resolutions, which clearly introduced biases especially in the subsurface. Also the impact of even more detailed parametrizations can easily be tested. E.g. one could alternatively use the tiling approach possible in CLM with several PFTs in one grid cell instead of only the dominant one as we did, or just a different microphysics scheme in the atmospheric component. Most probably only limited and well selected time periods would need to be re-run for such analyses.

In the future our setup can be extended to an ensemble. The data provided with our data set can be easily used to produce such an ensemble. E.g. the methods used to generate the soil and sub-soil can be used to produce a set of equally likely soil configurations given the always limited observations while atmospheric variability and uncertainty can be generated by the use of analysis ensembles for initialization and lateral boundary conditions.

All the points we mentioned here would be added to the manuscript in a slightly truncated form to make clear what kind of options are available with this dataset. We hope that this will inspire other people to use it rather than develop a setup on their own if possible.

---

## Author Response (AR1)

Changes made in reply to reviewer comments:

Below we detail the changes made to the manuscript in response to the reviewer comments:

**General remarks**:

1.) "*The applicability and purpose of the dataset is not entirely clear to me. The authors only briefly comment on this in the conclusions section, stating it could be used for exploring land-atmosphere feedbacks, investigating potential model simplifications, and data assimilation testing. Honestly, I do not see how the dataset can be useful in such analyses as (i) land-atmosphere interactions are known to be hardly robustly captured by models in general, and (ii) the testing of model simplifications and data assimilation would require different versions of the dataset with respective different model configurations in my opinion. In this context, I ask the authors to expand and clarify their discussion on the applicability of this dataset.*" -Reviewer 1

"*However, I still am left with the question of what is this data going to be used for beyond the work done by the authors. From my understanding, a publication in ESSD should be a dataset that one could expect to be used extensively by the larger community. If this covered the entire country or Europe then that would be a different story, but I just don't see it as is. I still believe the work should be published but the paper would greatly benefit by putting it all in a greater context. I suggest the discussion provide a much more detailed overview of what this data could be used for and why it is necessary to have a unique dataset instead of just having each researcher rerun the simulations. In essence, it would be nice to know what this dataset could provide the larger scientific community 5-10 years from now.*" -Reviewer 2

Both these raise similar points. We have changed the discussion section extensively, adding two paragraphs detailing further use of this dataset. For more details check our responses to the reviewers.

2.) "*CLM3.5 is used here as land surface model. This is outdated. By now, CLM5 is clearly more advanced in terms of simulating processes related to vegetation and hydrological dynamics at the land surface (Lawrence et al. 2019).*"

We added a paragraph where we explain that using a more up-to-date version would not have had in impact on this simulation due to our setup choices: "Version 3.5 of CLM that is used here is already relatively old. Even though version 5 was not yet available when we started our work, it is now and comparison is warranted. Newer versions of CLM have several major improvements over 3.5. The first one is a more sophisticated routing scheme leading to much improved soil moisture profiles. In our case we replace this part with ParFlow anyway so our older version is not a disadvantage in that regard. Other improvements are the inclusion of carbon and nitrogen cycles as well as more options for crop type vegetation. Here we purposely simplify our setup as we not only have and want static land use but also use a blend type of crop with no sharp changes in LAI due to harvests. Instead, we assume harvest to be an ongoing process all throughout autumn. Thus, all these improvements do not downgrade the simulation results presented and discussed in this study."

3.) "*Throughout the manuscript there are various inaccurate statements limiting the reproducibility of the model simulation (e.g. 'increased by about', 'increased by approximately', 'set considerably higher',*

*'needed to be increased from its standard values', various pedotransfer functions used without explaining criteria, see also respective specific comments below). More information needs to be provided in each of these cases to ensure the reproducibility of the entire analysis, either in the manuscript or in an appendix."*

We have changed the appropriate sections to be specific. The details of each change are given in the section below. Since the forcing and namelist files are part of the dataset as supplementary files, the exact setup of our models can be re-created and values checked. We added this information with a link to the forcing files to the text.

4.) *"There are several arbitrary choices made throughout the study which need to be (better) motivated. This includes modifications of the modeling setup of which the purpose is not clear or the magnitude is arbitrary (e.g. 20% increase of sand fraction, ignoring of karst layers, conceptualization of alluvial layers as gravel and bedrock layers including the assumption(s) of values for various involved parameters, modifications of the LAI data). When making these modifications to adapt the model behavior in particular respects (more sandy soils to enhance infiltration) it should be kept in mind that even if the particular purpose is fulfilled, the land-atmosphere system is highly interconnected such that unforeseen side effects can occur. Further, the arbitrary choices include the approach(es) used to validate the modeled dataset (e.g. spatial averaging of model data across 25 grid cells for validation of atmospheric boundary layer characteristics, seemingly random time intervals of the soil moisture, evapotranspiration and runoff validations)."*

We have clarified our reasons as to why we changed some of the values. Most are based on other studies or previous results. Again, for more details as to the nature of these changes please check the detailed response we gave.

5.) "Soil types are an important ingredient for hydro-climatological model simulations. The downscaling-based derivation of soil types in this study is (i) difficult to understand and (ii) contains several assumptions which are not motivated, among which are the amount of considered 1995 locations, the 20% increase of sand fraction (see above), the choice of an exponential model, the choice of conditional co-simulation versus kriging, the focus on first three soil horizons (first means uppermost I guess?). I wonder what is the impact of the choices made here on the final dataset?"

I hope we have sufficiently answered this in our comment. We still have updated the related section to be clearer regarding the process and the impact of our choices. We have some more details below where they are mentioned again.

6.) *"The validation of the model simulation in terms of evapotranspiration is very limited. While it is reassuring that the ET and groundwater dynamics are broadly coupled according to expectations this is not a quantitative assessment. The modeled ET could instead (or additionally) be compared with state-of-the-art evapotranspiration datasets such as GLEAM (Martens et al. 2017) or FLUXCOM (Jung et al. 2019) at larger spatial scales."*

We have added and additional paragraph in detailing the reason for this very limited analysis. The various reasons are detailed in our response. The change is this: "We want to point out that in this region ET is almost always limited by atmospheric demand which is why we limit his analysis to bare-soil and evaporation only. Since the upper-most layers can dry quickly the resulting drop in evaporation can

be seen which is not the case for ET if there is an extended root zone as we have even for crop and grassland. These bare-soil areas are not a feature of the real catchment and as such cannot be compared to real measurements."

7.) "I like the comprehensive validation of the dataset in terms of several variables - an overview table summarizing the determined strengths and weaknesses would be helpful for users I think."

We have added such a table as table A.3.

8.) "There are too many figures in my opinion, diluting the main messages. Figures 3-5, 7, 14, 17 could be moved to supplementary, and Figures 9 & 10 could be combined."

We have removed former figure 7 and moved former figures 2,4,5,14 and 17 to the appendix. In addition, we updated several of them in accordance with the more specific comments below.

**Specific comments**:

"*line 35 and throughout: 'simulated' would be more straightforward than 'virtual', usingsuch terminology the term 'real' (line 34 and throughout), referring to observations, canbe removed from the manuscript*"

We have used simulated instead of virtual throughout the text now.

"*line 56: test a disaggregation method*"

In the cited paper this was done for soil moisture. Doing this for this coupled system and all related variables and parameters would be a complete study on its own. The paper was cited to highlight the uses of simulated datasets rather as an example we wanted to follow.

"*line 75-78: you do not aim to reproduce to observed catchment dynamics but still validate the model in some respects - this seems contradictory to me; what is the aim here if not validating the model against observations? how useful is a modeled dataset for the community if is not resembling observations?*"

We have added a sentence in section 4 explaining why we validate our simulation even if we have purposely deviated from reality in some cases: "Even though we do not aim to be as close to reality as possible, we feel it important to show that the model system is behaving as expected and is thus suitable for the various use cases we discussed."

"*lines 139 & 145: the chosen time period and simulation catchment/area are not motivated*"

We have added a sentence to detail that the time period limit wat simply due to forcing availability. The choice of the catchment was arbitrary and it was only chosen because of size and because it fits in a roughly square computational domain: "... as 2007 was the first full year where high resolution atmospheric forcing was available and nine years was the maximum possible length with our granted compute time."

"*line 172: please give more information on the 'software restriction'*"

The restriction was somewhat unique with the system we were running on. We have added this information but it will likely not be useful any more as the system (and other like it) have been replaced over the last couple years and this restriction no longer applies.

*"line 183: please give more information on the location of the grid cells and the artificial elevation modification to ensure reproducibility, here or in an appendix"*

We have added a remark linking to the provided forcing data where the full elevation map is available: "All these changes are part of the forcing files that are provided with the full dataset making it easy to reproduce our simulations (https://cera-www.dkrz.de/WDCC/ui/cerasearch/entry?acronym=Neckar_VCS_v1_FORCING)"

*"lines 195-196: 'about 20%' & 'about 3.3', please be more accurate"*

We have clarified that 20% is an area average depending on PFT. The 3.3 change is due to the general LAI value of needle-leaf trees in that region which we calrified as well.

*"lines 194-197: in the abstract of the Tian et al. 2004 paper I found "On average, themodel [...] overestimates FPAR over most areas in the Northern Hemisphere com- pared to MODIS observations during all seasons except northern middle latitude summer." "The MODIS LAI is generally consistent with the model during the snow-free periods..." which makes me wonder why the authors modify LAI in summer? Further this could create jumps in the LAI time series from May-June and August-September. More importantly, you state here that LAI is used "for the year 2008". Does this mean there is no interannual vegetation variability? This would affect evapotranspiration and thereby many related variables and would need to be stated as a serious shortcoming."*

We have added several lines detailing that we do indeed ignore interannual variability on purpose and why (again see our detailed response): We also clarified that the changes in summer are small and related to the aggregation of shrubs and proper forests and that there are no jumps for LAI possible due to the way CLM treats LAI: "As a result, interannual variability is not considered in this simulation as we have the same LAI curve for each PFT each year. While this somewhat limits the comparability to ET observations in spring it is somewhat lessend by the fact that CLM does always use the values from two months and interpolates based on the date (for instance on the 1st of April the values from March and April would have almost equal weight, while on the 14th of April the April weight is clearly dominant)."

*"line 204: please give the spatial resolution instead of the scale 1:1000000"*

We added this. (~1km, not constant due to map projection)

*"line 221: 'approximately 20%', please be more accurate; further, and more importantly, please motivate this modification and its magnitude"*

We have added that is 20% except for cases where sand% was already very high to avoid unrealistic cases of >90% sand. We clarified that this is due to earlier simulation results and the resolution that we use: "...increased by 20% (for clay-rich areas that previously had practically no sand, less for areas that already had high sand content to avoid cases with >90% sand) resulting in a slightly higher hydraulic conductivity because previous simulations yielded too shallow unsaturated zones as was expected for the resoltion of this simulation. The change to sand content was a direct response to this issue as it fixed most of the emerging biases"

*"lines 229-231: please give detailed information on where which pedotransfer functions have been used"*

We have added more detail, the text now reads: "The pedotransfer functions of Cosby et al. (1984) is used to estimate saturated hydraulic conductivity based on soil texture, the one from Rawls (1983) is used to estimate soil bulk density based on soil texture and organic matter and the one from Tóth et al. (2015) is used to estimate van Genucthen parameters based on soil texture and bulk density. These have been selected based on data availability, applicability of the particular approaches, and previous evaluations conducted in the area (Tietje and Hennings, 1996)."

*"lines 239 & 242: repetition of the information that karst is not considered"*

Mentioned it only the first time.

*"lines 239-240: 'to avoid the manifold hydrological challenges related to its modeling', please be more specific here, also please comment on the impact of this simplification of the approach on the final dataset"*

As mentioned in our detailed reply, karst is very hard to model correctly, especially with only 100m of depth of a model. Because of that we just disregard it. We added a sentence to make clear that while overall accuracy of groundwater suffers, effects on the near-surface soil moisture in these areas is limited: "While this can have significant impact on groundwater equilibrium and dynamics on the decade time scale, for the rather short time period considered here we would simply be left with a bias."

*"line 245: if these alluvial bodies are so relevant, why does this study use datasets which do not include them?"*

The datasets we used are the most detailed ones for that region and even those do not include them for smaller rivers. They are very important for accurate subsurface flow which is why we added them. We made it clear in the text that they are not resolved in the dataset: "(not part of the soil and not large enough to be resolved in the geological map)"

*"line 249: evapotranspiration errors in models can be significant and might be underestimated here"*

We already assume a 30% error here and because we are energy limited this is very conservative: "...as in this climate we are almost always energy limited and therefore ET differences are reduced to LAI differences"

*"section 4: please discuss for the performed validation analyses how the determined performance of the study dataset compares generally with the performance of other regional climate models im similar hydro-climatic regions"*

As mentioned in our response, the goal of this study is not to be in line with climate models. Indeed, we show a more static picture, purposely ignoring climate change and accompanying effects to concentrate on compartment interaction instead. We added in the discussion the possibility for investigating this as a separate piece of work: "One of these topics is considering more transient changes (LAI and climate change) rather than the more static picture we present."

*"line 277: 'simulated realistically', please give more details here on how this is quantified"*

Added more detail: "(timing, strength of wind gusts, change of wind direction, change in temperature and pressure)"

*"line 301: 1km2, is this referring to the spatial resolution being 1km x 1km?"*

It is 1x1km, made this clear in the text.

*"line 323: 'quite well', please be more specific and objective. Further, in Figure 9a between 6h-17h the pattern in the model is actually opposite to that of the observations, I would not refer to this as fitting "quite well"."*

We have added a section explaining this in more detail and pointing out that afternoon rain is underestimated, which is normal with the model we use: "especially in late afternoon and night while it overestimates precipitation during the late morning while underestimating it in early afternoon in summer. In Winter this effect is much less pronounced but still there. It is connected to convective showers that are still too small and parametrized. These parametrizations were not originally designed for the km scale and have issues at this resolution and start producing precipitation too early."

*"line 331: the potential (dis)agreement of simulated and actual land cover can be checked using high-resolution land cover datasets such as provided by ESA CCI line 343: how are the temperature standard deviations determined?"*

While the mismatch can be checked we still have to account for it when comparing results. We made this clear in the text. We also explained how the temperature deviations are calculated.

*"line 361: 'very well', please be more specific and objective"*

Changed wording and added more detail: "... are very close to observations during the day and at heights above 10m."

*"line 367: why not using the ESA CCI soil moisture dataset derived from observations of various satellites for this validation?"*

Since we different land-use and soil properties a comparison would not give any conclusive result.

*"lines 394-395: I do not really understand why this daily matching is applied here? Also it is not clear how this is done."*

This means that for each day a factor is calculated and the actual result is corrected by this factor: "...and applied as a correction factor."

*"lines 391 & 396: I guess you are referring to Figure 15 here, not Figure 16 as stated."*

Figures have changed greatly, all figures should now be referenced correctly.

"line 415: 'adequate agreement', please be more specific and objective"

Changed wording: "... similar to the observations showing the same responses to rain events as well as similar behavior during dry periods, which is noteworthy...

"line 422: 'will always be replaced' needs to be toned down in my opinion"

We now say "often" instead of "always".

"line 440: 'good distribution', please be more specific and objective"

Changed line: "...reasonable split between shallower and deeper (5 meter and below) groundwater tables compared to expected values from observations with shallower levels overall."

"lines 447-450: I do not understand how the "fluctuations" are "scaled". Do you divide by the inter-annual standard deviation to obtain normalized anomalies (or z-scores)? If so, please name it this way as the term "fluctuations" is rather unclear."

In this case fluctuation merely means that the yearly average is removed for both time series: "(subtracting the yearly mean from the respective time series)"

"line 450: I guess this should be "according to Figure 19b" and not 19c?"

Again, figures have changed, should be correct now.

"lines 451-452: How is this trend computed?"

This was poor wording. We changed it to "fluctuations" to indicate that the same time-series as before is referenced.

"line 454 and following: I like this discussion of limitations and issues"

Thanks.

"line 458: to me it seems three challenges being discussed here (?)"

This was an error left over from a previous version. It is corrected.

"lines 502-512: As there are multiple concrete ideas to improve the model setup and consequently the dataset, why not implementing them before publishing this dataset? comment if this will be game-changers"

We have added a comment giving our opinion on this: "While these changes would show improvements, they are likely marginal or very specific (river discharge characteristics) and would therefore not warrant the great computational cost to re-run for such a long time. Future model developments of TerrSysMP may enable this option and it would be interesting to compare resulting datasets and quantify the simulation speed increase by using GPU compute technologies."

"Figure 2: This figure is the same as in the Gasper et al. 2014 paper, with the reference given in the caption. I think it is uncommon to use figures from previous papers, so I would remove this and only refer to the figure in the reference paper in the main text."

Moved to Appendix

"Figure 3: Maybe I missed that but what is domain 1?"

Again, left over from an older version, it is now simply the "simulated domain"

"Figure 4: "e+00" can be removed"

removed

"Figures 4-6: Please label the color bars."

done

"Figure 6: Please harmonize "evaporation" and "evapotranspiration" in the caption and the axis label. The same applies for the main text in section 4.1."

We changed everything to "evaporation" in the text and figures.

"Figure 8: Values are quite far apart from color bars. Also, it would be nice to also express the difference as percentage."

We removed the large spaces between values. A percentage difference would show almost the same picture, just with lower values in the mountains. We therefore left this out.

"Figure 11: y-axis label missing Please explain what is meant with the "temperature standard deviations""

Added label and explained standard deviations in the text: "(mean absolute difference between daily and monthly mean profile)"

"Figure 13: Please specify from which times the reference radiosonde observations are taken. Further, please explain how the standard deviation is derived."

Same as last figure, also the times are in the figure already. Also added them to the caption.

"Figure 15: It would be insightful to quantify the agreement of the temporal dynamics with e.g. a correlation."

This option is discussed in the text instead.

"Figures 16-18: Please use the station names throughout instead of the position numbers."

Changed to station names.

"Figure 19: Panel a is not labelled, as well as color bar and axes therein The terms "model" and "reality" are not consistent with the terminology used throughout the manuscript."

Changed the figure accordingly.

---

## Author Response (AR2)

Changes made in reply to reviewer comments:

Below we detail the changes made to the manuscript in response to the reviewer comments:

**Reviewer 1:**

**General remarks**:

1.a) "*The applicability and purpose of the dataset is not entirely clear to me. The authors only briefly comment on this in the conclusions section, stating it could be used for exploring land-atmosphere feedbacks, investigating potential model simplifications, and data assimilation testing. Honestly, I do not see how the dataset can be useful in such analyses as (i) land-atmosphere interactions are known to be hardly robustly captured by models in general, and (ii) the testing of model simplifications and data assimilation would require different versions of the dataset with respective different model configurations in my opinion. In this context, I ask the authors to expand and clarify their discussion on the applicability of this dataset.*"

*Here we want to state again the response we originally supplied with the answer to the first review:*

"Indeed, we agree with the reviewer that the purpose of this dataset is a key characteristic that should be clear from the manuscript. Within our project, we are working on developing and testing data assimilation approaches. More specifically, we aim to disentangle the value of different observation types in fully coupled models. We used the fully coupled model to develop a virtual reality. We faced two important issues. First, the reliability of the created virtual reality. Second, the huge computational effort needed to run a simulation with the desired spatial and temporal resolution. We decided to work therefore on this study to 1) provide a comprehensive assessment of the reliability of the states and fluxes simulated by the fully coupled models, and 2) make not only model code, forcings and parameters available but also the complete model output to guarantee the reproducibility of the entire model. This should allow the use of this model setup by other groups to do targeted tests, evaluating for example the impact of a different spatial model resolution, or a specific simplification (for example, neglecting lateral flows in the subsurface). This is challenging because we aimed at mimicking the real world without doing a formal model calibration. We believe that the comparison with real observations within the area supports the capability of the fully coupled model to reproduce the water and energy cycles and should provide the confidence to use the dataset.

The dataset will be used for data assimilation experiments in our project, but is available for anybody interested in data assimilation and modelling experiments. We will extract (virtual) observations from the virtual reality and perturb the observations with a measurement error. The effectiveness of any data assimilation algorithm can be quantified making it a powerful tool for the development of such algorithms. In addition, thanks to the availability of a fully coupled system, we will be able to check the effect of assimilating observations in the different terrestrial compartments (e.g., assimilating groundwater level data and check effect on land-atmosphere fluxes like evapotranspiration).

In addition to this primary purpose, this dataset can be exploited for model simplification, to test reconstruction methods or define monitoring networks. Examples from our work are listed:

Baroni, Gabriele, Bernd Schalge, Oldrich Rakovec, Rohini Kumar, Lennart Schüler, Luis Samaniego, Clemens Simmer, and Sabine Attinger. "A Comprehensive Distributed Hydrological Modelling Inter-

Comparison to Support Processes Representation and Data Collection Strategies." Water Resources Research, January 17, 2019. https://doi.org/10.1029/2018WR023941.

Lv, Shaoning, Bernd Schalge, Pablo Saavedra Garfias, and Clemens Simmer. "Required Sampling Density of Ground-Based Soil Moisture and Brightness Temperature Observations for Calibration and Validation of L-Band Satellite Observations Based on a Virtual Reality." Hydrology and Earth System Sciences 24, no. 4 (April 17, 2020): 1957–73. https://doi.org/10.5194/hess-24-1957-2020.

 Haese, B., S. Hörning, C. Chwala, A. Bárdossy, B. Schalge, and H. Kunstmann. "Stochastic Reconstruction and Interpolation of Precipitation Fields Using Combined Information of Commercial Microwave Links and Rain Gauges." Water Resources Research 53, no. 12 (2017): 10740–56. https://doi.org/10.1002/2017WR021015.

In Baroni et al. (2019) a similar dataset created with TerrSysMP is compared to a distributed, conceptual-based hydrological model. The comparison was able to identify where and when simulations yielded similar results (and as such the simplified model could be used instead of the complex one) and where differences emerged (and as such identifying the need of model improvements and measurements). In Lv et al. (2020) the virtual reality was used to identify optimal sampling strategies for capturing soil moisture dynamics at a desired accuracy and precision. This is a basis for designing cost-efficient monitoring schemes. Haese et al. (2017) introduced a method to reconstruct precipitation fields combining precipitation measured by rain gauges and commercial microwave links. The synthetic dataset allowed the comparison of the reconstructed fields and its estimated error against the virtual truth. These are just few examples of how the data set can be used to test different methods or modeling strategies.

In addition, it is important to point out that indeed land-atmosphere feedbacks are difficult to capture by models. The advantage of our coupled modelling approach is that we replace the hydrology of the land surface model CLM3.5 with an approach where lateral subsurface flows (soil and, especially important, groundwater) are considered as well, and also lateral flows along the land surface (streams). As such, it is a state of the art model for representing land-atmosphere interactions, especially for the water and energy cycles. We agree that using CLM3.5 instead of CLM5.0 implies that the representation of vegetation and biogeochemical cycles in land-atmosphere interaction it is not the best we can do at the moment. However, a coupled land-atmosphere model using CLM5.0 and at the same time a full subsurface hydrology including lateral flows is not available yet.*

Based on this response we have added a paragraph in the discussion section (lines 589-609):

 "Finally, we want to address the applicability and usefulness of this dataset for various studies. As indicated, this dataset can be valuable for data assimilation both for testing new methods or algorithms and as a standard set for synthetic observations to pull from. It is thus possible to carry out data assimilation experiments with different conditioning datasets. Due to the long time series we have covered almost any possible weather regime (with the exception of truly extreme events) which can be a great advantage as some algorithms may work well for most conditions but may show weaknesses for other specific conditions (for instance the CMEM operator in combination with frozen soils). It also allows to investigate the impact of simplifications such as using a fixed atmospheric forcing instead of a model and thus disregarding feedback mechanisms. Next to data assimilation there are also model development and model analysis and comparison studies that can benefit from this dataset. If specific changes to the

model system are made, for example testing a new cloud parametrization, all of the input files that are provided with this dataset can be used to quickly set up a working environment with known results to compare to. Here the length of the simulation is again an advantage since any development can be tested for relevant time slices. A detailed analysis of the dataset regarding compartment interactions is also of interest. We have shown the overall behavior of the system but we have not studied specific interesting events such as heatwaves, dry periods or floods in detail. It would also be of interest to perform longer term simulations to analyze climate change and analyze better inter-annual variability by considering yearly changes in the LAI cycle. Lastly, this setup can also be considered as a template for ensemble-based setups in the future. Right now, reduced resolutions are needed in order to run many members of such a coupled model system. As we have shown, even this higher resolved simulation still shows some biases that are directly related to resolution so increasing resolution also in ensembles will be a logical step in the future to obtain better results. When this happens, the methods we used here to generate this simulation will be very useful as well as the analysis presented here to decide how an ensemble should be set up based on the goal (an ensemble for flood forecasts would benefit from a different strategy than an ensemble for drought monitoring)."

2.) "*CLM3.5 is used here as land surface model. This is outdated. By now, CLM5 is clearly more advanced in terms of simulating processes related to vegetation and hydrological dynamics at the land surface (Lawrence et al. 2019).*"

We added a paragraph where we explain that using a more up-to-date version would not have had an impact on this simulation due to our setup choices (lines 123-131):

"Version 3.5 of CLM that is used here is already relatively old. Even though version 5 was not yet available when we started our work, it is now and comparison is warranted. Newer versions of CLM have several major improvements over 3.5. The first one is a more sophisticated routing scheme leading to much improved soil moisture profiles. In our case we replace this part with ParFlow anyway so our older version is not a disadvantage in that regard. Other improvements are the inclusion of carbon and nitrogen cycles as well as more options for crop type vegetation. Here we purposely simplify our setup as we not only have and want static land use but also use a blend type of crop with no sharp changes in LAI due to harvests. Instead, we assume harvest to be an ongoing process all throughout autumn. Thus, all these improvements do not downgrade the simulation results presented and discussed in this study."

3.) "*Throughout the manuscript there are various inaccurate statements limiting the reproducibility of the model simulation (e.g. 'increased by about', 'increased by approximately', 'set considerably higher', 'needed to be increased from its standard values', various pedotransfer functions used without explaining criteria, see also respective specific comments below). More information needs to be provided in each of these cases to ensure the reproducibility of the entire analysis, either in the manuscript or in an appendix.*"

We have changed the appropriate sections to be specific. The detailed changes are given in the specific answers to reviewer questions below. Since the forcing and namelist files are part of the dataset as supplementary files, the exact setup of our models can be re-created and values checked. We added a direct link to where the forcing files can be downloaded (lines 269-270):

"All these changes are part of the forcing files that are provided with the full dataset making it easy to reproduce our simulations (https://cera-www.dkrz.de/WDCC/ui/cerasearch/entry?acronym=Neckar_VCS_v1_FORCING)"

4.) "*There are several arbitrary choices made throughout the study which need to be (better) motivated. This includes modifications of the modeling setup of which the purpose is not clear or the magnitude is arbitrary (e.g. 20% increase of sand fraction, ignoring of karst layers, conceptualization of alluvial layers as gravel and bedrock layers including the assumption(s) of values for various involved parameters, modifications of the LAI data). When making these modifications to adapt the model behavior in particular respects (more sandy soils to enhance infiltration) it should be kept in mind that even if the particular purpose is fulfilled, the land-atmosphere system is highly interconnected such that unforeseen side effects can occur. Further, the arbitrary choices include the approach(es) used to validate the modeled dataset (e.g. spatial averaging of model data across 25 grid cells for validation of atmospheric boundary layer characteristics, seemingly random time intervals of the soil moisture, evapotranspiration and runoff validations).*"

We have clarified our reasons as to why we changed some of the values. Most are based on other (cited) studies, previous results or the specific setup and simplifications we used. All these points are handled separately in the detailed section below.

5.) "Soil types are an important ingredient for hydro-climatological model simulations. The downscaling-based derivation of soil types in this study is (i) difficult to understand and (ii) contains several assumptions which are not motivated, among which are the amount of considered 1995 locations, the 20% increase of sand fraction (see above), the choice of an exponential model, the choice of conditional co-simulation versus kriging, the focus on first three soil horizons (first means uppermost I guess?). I wonder what is the impact of the choices made here on the final dataset?"

The choice for the increase in sand content was based on earlier results where we used this setup in a sub-catchment (see images below) and tested the response to various soil setups. We found there that we needed to increase sand content by 20% as we otherwise would have had very shallow water-tables and even higher ET values. The changed text is given below responding to the specific comment originally at line 221.

As for the process that we used to create the soil map, we want to repeat our earlier comment from the response to the first reviewer on this:

"Indeed, the method to downscale the original soil map underwent several tests. Some of them have been described in (Baroni et al., 2017). We have referred to that paper also within the manuscript, but we summarize below the main steps for the sake of clarity.

The original soil map with necessary data for running the model is quite coarse with the areas of most of the soil polygons above 20 km2 (Figure 1).

[Figure]

Figure 1: original soil map of the area. Please note that these figures do not represent the entire domain of the VR1 but they are reported to show the main characteristics.

Based on that, different methods have been tested and compared to the new so-called conditional point method (CPM). The results are depicted in Figure 2.

[Figure]

Figure 2: soil clay map created with different methods. From left: random error (RE) method where nominal value of each soil type has been perturbed, spatially correlated (SC) method where a spatial random variable has been superimposed to the original soil map (black line) and, conditional point (CP) method where conditional points are extracted and used in the conditional simulation.

The results showed that the CP method introduces uncertainty only at small spatial scales while the longer spatial patterns are preserved without sharp changes between the soil units. For this reason, this method has also been selected within the present study. Please also note that we did not consider the kriging approach as kriging removes small-scale variability which was considered an important feature to be considered in the dataset and coupled modelling.

The conditional point CP method is described in the next Figure 3.

[Figure]

Figure 3: sketch of the method to downscale the soil map (a transect is depicted as example)

Starting from the original soil map, (step C1) soil samples are selected, (step C2) a variogram model is fitted and (step C3) a conditional simulation is created. In total 1995 locations have been used to mimic a realistic number of soil samples that can be collected for creating a soil map. More specifically a density of 1 sample per 5 km2 is used. Some tests conducted for different sampling densities did not change the results significantly. The procedure will be discussed in greater detail in the new version of the manuscript. In contrast, the selection of the variogram model did not undergo a detailed analysis but was selected to provide a stronger short scale variability in contrast to the Gaussian variogram. Finally, the use of the first upper soil horizons has been considered because they represent the main soil horizons of the root zone (horizons A, B and C). Additionally, the use of only the first three horizons allowed us to have a variable soil depth under which we imposed the rocks. The original soil map provides more soil layers which would have resulted in a uniform soil depth of 2m.

As previously discussed, all these features have been selected after several tests conducted with the fully coupled model or with simpler configurations (not coupled, short period, limited areas), finally resulting in more realistic dynamics. We did not completely evaluate the impact of all these decisions on this dataset. This would require a detailed sensitivity analysis that is beyond the scope of the present study and also not feasible from the computational point of view."

Based on this response we have updated the entire section detailing the creation of the soil map in various places. It now reads (lines 238-263):

"Since soil properties may vary substantially at scales smaller than the 1km for which BUEK1000 is appropriate, which might impact system dynamics (Binley et al. 1989, Herbst et al. 2006, Rawls 1983), the soil map is downscaled by artificially adding variability using the conditional points method recently presented in Baroni et al. (2017) as follows:

(1)     The BUEK1000 soil map is randomly sampled at 1995 point locations with one sample every 5 km$^2$ on average, a minimum sample distance of 250 m, and at least one sample for each soil type of the original soil map which is realistic in the context of how soil maps are usually created. This strategy resulted from extensive testing by minimizing the tradeoffs between reproducing the main features of the original soil map and creating variability at finer resolution.

(2)     The sample locations are used as conditional points for further interpolation. Here, texture, carbon content, and depth of the first three soil horizons are extracted from the BUEK1000 resulting in variable soil depth rather than the assumed unrealistic uniform soil depth. In addition, the sand content of the original map was increased by 20% (except for  areas with very high sand content to avoid grid cells with >90% sand) resulting in a slightly higher hydraulic conductivity because previous simulations yielded too shallow unsaturated zones related to the spatial resolution of the simulation. Changing sand content increased the thickness of unsaturated zones and lowered groundwater tables, fixing most of the emerging biases

(3)     Experimental variograms and cross-variograms are calculated for all variables and exponential models were fitted to all spatial structures.

(4)    A texture map (sand and clay percentage) is generated using a single realization based on conditional co-simulation (Gomez-Hernandez and Journal, 1993) to provide the sub-scale variability (<1 km$^2$). Soil horizon depths and carbon content are, however, assumed to have a smoothed spatial variability; therefore, they are interpolated based on ordinary kriging as the removal of small-scale variability is not important for the depth and carbon content.

(5)    Since ParFlow describes retention and hydraulic conductivity curves based on Mualem-van-Genuchten parameters, pedotransfer functions are applied to estimate these parameters. The pedotransfer function of Cosby et al. (1984) is used to estimate saturated hydraulic conductivity based on soil texture, the one from Rawls (1983) is used to estimate soil bulk density based on soil texture and organic matter and the one from Tóth et al. (2015) is used to estimate van Genuchten parameters based on soil texture and bulk density. These have been selected based on data availability, applicability of the particular approaches, and previous evaluations conducted in the area (Tietje and Hennings, 1996)."

6.) "*The validation of the model simulation in terms of evapotranspiration is very limited. While it is reassuring that the ET and groundwater dynamics are broadly coupled according to expectations this is not a quantitative assessment. The modeled ET could instead (or additionally) be compared with state-of-the-art evapotranspiration datasets such as GLEAM (Martens et al. 2017) or FLUXCOM (Jung et al. 2019) at larger spatial scales.*"

We have added a paragraph detailing the reason for this very limited analysis. Several of our setup choices (no interannual variability and one land-use type per gridcell) make it hard to compare to observational datasets. ET is very sensitive to land-use and we use a different land-use map than either of these datasets. We also use bare-soil instead of urban areas so given the resolution of these datasets and the rather high density of urban areas in the region emerging differences would be hard to attribute. The updated text now includes this information (lines 347-350):

"We want to point out that in this region ET is almost always limited by atmospheric demand which is why we limit the analysis to bare-soil evaporation only. Since the upper-most layers can dry quickly the resulting drop in evaporation can be seen which is not the case for ET if there is an extended root zone as we have for crop, grassland and forests. These bare-soil areas are not a feature of the real catchment and as such cannot be compared to real measurements."

7.) "I like the comprehensive validation of the dataset in terms of several variables - an overview table summarizing the determined strengths and weaknesses would be helpful for users I think."

We have added such a table, see Table A.3.

8.) "There are too many figures in my opinion, diluting the main messages. Figures 3-5, 7, 14, 17 could be moved to supplementary, and Figures 9 & 10 could be combined."

We have removed former figure 7 and moved former figures 2, 4, 5, 14 and 17 to the appendix. In addition, we updated several of them in accordance with the more specific comments below.

**Specific comments**:

*"line 35 and throughout: 'simulated' would be more straightforward than 'virtual', using such terminology the term 'real' (line 34 and throughout), referring to observations, can be removed from the manuscript"*

We have used simulated instead of virtual throughout the text now.

*"line 56: test a disaggregation method"*

In the cited paper this was done for soil moisture. Doing this for this coupled system and all related variables and parameters would be a complete study on its own. The paper was cited to highlight the use of simulated datasets rather as an example we wanted to follow.

*"line 75-78: you do not aim to reproduce to observed catchment dynamics but still validate the model in some respects - this seems contradictory to me; what is the aim here if not validating the model against observations? how useful is a modeled dataset for the community if is not resembling observations?"*

We have added a sentence in section 4 explaining why we validate our simulation even if we have purposely deviated from reality in some cases (lines 315-316):

"Even though we do not aim to be as close to reality as possible, we feel that it is important to show that the model system is behaving as expected and is thus suitable for the various use cases we discussed."

*"lines 139 & 145: the chosen time period and simulation catchment/area are not motivated"*

We have added a sentence to detail that the time period limit was simply due to the availability of atmospheric forcings. The choice of the catchment was driven by water-table depth. Given the shallow groundwater tables in large part of the catchment a stronger feedback of groundwater on atmospheric conditions can be expected:

"We ran the fully-coupled model for a period of nine years (2007-2015) as 2007 was the first full year where high resolution atmospheric forcings were available and nine years was the maximum possible simulation length given constraints on compute resources." (line 154-156)

"These typical central European catchment features in addition to the relatively shallow groundwater tables (implying a stronger possible feedback of groundwater on atmospheric conditions) were the basis to select the Neckar catchment for our simulation." (lines 174-176)

*"line 172: please give more information on the 'software restriction'"*

The restriction was somewhat unique related to the system we were using for our runs. We added the following text (lines 190-193):

 "A software restriction (unfixable bug specific to the supercomputing system we were using for our simulation runs as described in the previous section) did not allow for cases with more than 4.2 million CLM columns as the model did not initialize properly and crashed implying that a higher spatial resolution for CLM and ParFlow than 400 m could not be achieved for the Neckar catchment on the used system."

*"line 183: please give more information on the location of the grid cells and the artificial elevation modification to ensure reproducibility, here or in an appendix"*

We have added a remark linking to the provided forcing data where the full elevation map is available (lines 269-270):

"All these changes are part of the forcing files that are provided with the full dataset making it easy to reproduce our simulations (https://cera-www.dkrz.de/WDCC/ui/cerasearch/entry?acronym=Neckar_VCS_v1_FORCING)"

*"lines 195-196: 'about 20%' & 'about 3.3', please be more accurate"*

In order to handle the reviewer comments, we modified the text as follows (lines 218-223):

"This LAI is increased for all plant functional types by 20 percent on average (more for forests and less for grassland and crops) in the summer months. LAI was changed by factors less than 1 to 3.3 in winter-time (DJF average) for needle-leaf forests in order to account for known biases in the MODIS data (Tian et al. 2004) mostly related to snow cover and fractional land cover due to the satellite footprint which often includes other vegetation types or roads and other buildings, leading to an underestimation for a gridcell that is fully covered by just one type as we use them."

*"lines 194-197: in the abstract of the Tian et al. 2004 paper I found "On average, the model [...] overestimates FPAR over most areas in the Northern Hemisphere compared to MODIS observations during all seasons except northern middle latitude summer." "The MODIS LAI is generally consistent with the model during the snow-free periods..." which makes me wonder why the authors modify LAI in summer? Further this could create jumps in the LAI time series from May-June and August-September. More importantly, you state here that LAI is used "for the year 2008". Does this mean there is no interannual vegetation variability? This would affect evapotranspiration and thereby many related variables and would need to be stated as a serious shortcoming."*

We have added several lines detailing that we do indeed ignore interannual variability on purpose: We also clarified that the changes in summer are small and related to the aggregation of shrubs and proper forests and that there are no jumps for LAI possible due to the way CLM treats LAI (lines 216-217):

"As a result, interannual variability is not considered in this simulation and we have the same LAI curve for each PFT each year. This somewhat limits the comparability to ET observations especially in spring."

*"line 204: please give the spatial resolution instead of the scale 1:1000000"*

We added this. (~1km, not constant due to map projection)

*"line 221: 'approximately 20%', please be more accurate; further, and more importantly, please motivate this modification and its magnitude"*

We have added that it is 20% except for cases where sand% was already very high to avoid unrealistic cases of >90% sand. We clarified that this is due to earlier simulation results and the resolution that we use (lines 247-251):

"In addition, the sand content of the original map was increased by 20% (less for areas that already had high sand content to avoid cases with >90% sand) resulting in a slightly higher hydraulic conductivity because previous simulations yielded too shallow unsaturated zones as was expected for the resolution of this simulation. The increase of sand content fixed most of this bias."

*"lines 229-231: please give detailed information on where which pedotransfer functions have been used"*

We have added more detail, the text now reads (lines 259-263):

"The pedotransfer function of Cosby et al. (1984) is used to estimate saturated hydraulic conductivity based on soil texture, the one from Rawls (1983) is used to estimate soil bulk density based on soil texture and organic matter and the one from Tóth et al. (2015) is used to estimate van Genuchten parameters based on soil texture and bulk density. These have been selected based on data availability, applicability of the particular approaches, and previous evaluations conducted in the area (Tietje and Hennings, 1996)."

*"lines 239 & 242: repetition of the information that karst is not considered"*

Mentioned it only the first time.

*"lines 239-240: 'to avoid the manifold hydrological challenges related to its modeling', please be more specific here, also please comment on the impact of this simplification of the approach on the final dataset"*

As mentioned in our detailed reply, karst is very hard to model correctly, especially with only 100m model depth. Because of that we just disregard it. We added a sentence to make clear that while the overall representation of groundwater is poorer in these areas, effects on near-surface soil moisture in these areas is limited (lines 274-276):

"While this can have significant impact on the groundwater representation in the karst areas, for the rather short time period considered here we expect a limited impact on near-surface soil moisture content as the affected areas have in general deeper groundwater levels."

*"line 245: if these alluvial bodies are so relevant, why does this study use datasets which do not include them?"*

The datasets we used are the most detailed ones for that region and even those do not include them for smaller rivers. They are very important for accurate subsurface flow representation which is why we added them. We made it clear in the text that they are not resolved in the dataset (lines 281-282):

"Not covered by the discussed data sets (not part of the soil and not large enough to be resolved in the geological map) are the large alluvial bodies filling large part of the Neckar valley throughout the domain (Riva et al., 2006)."

*"line 249: evapotranspiration errors in models can be significant and might be underestimated here"*

In order to handle this comment we changed the text as follows (lines 285-288):

"While our simulated evapotranspiration rates may be inaccurate, it is implausible that this can account for 30% of the precipitation as in this climate we are almost always energy limited and therefore ET errors will be smaller and mostly related to errors in atmospheric forcings and LAI. This implies that the water could only have left the domain through the subsurface."

*"section 4: please discuss for the performed validation analyses how the determined performance of the study dataset compares generally with the performance of other regional climate models in similar hydro-climatic regions"*

As mentioned in our response, the goal of this study is not to be in line with regional climate models. We focus here on compartment interactions for a shorter time period where climate change is not the main focus. A further comparison with outcomes of regional climate models is beyond the scope of this work. We added in the discussion the possibility for investigating this as a separate piece of work (lines 600-603, part of the new paragraph included at the end of the discussion in reply to the first comment):

"We have shown the overall behavior of the system but we have not studied specific interesting events such as heatwaves, dry periods or floods in detail. It would also be of interest to perform longer term simulations to analyze climate change and analyze better inter-annual variabilities by considering yearly changes in the LAI cycle."

*"line 277: 'simulated realistically', please give more details here on how this is quantified"*

Added more detail: "(timing, strength of wind gusts, change of wind direction, change in air temperature and air pressure)"

*"line 301: 1km2, is this referring to the spatial resolution being 1km x 1km?"*

It is 1x1km, made this clear in the text.

*"line 323: 'quite well', please be more specific and objective. Further, in Figure 9a between 6h-17h the pattern in the model is actually opposite to that of the observations, I would not refer to this as fitting "quite well"."*

We have added a section explaining this in more detail and pointing out that afternoon rain is underestimated, which is a specific model feature (lines 374-378):

"The simulated daily precipitation distribution fits the observations especially in late afternoon and night while it overestimates precipitation during the late morning and underestimates it in early afternoon in summer. In winter this effect is much less pronounced . This behavior is related to the representation of convective showers in the atmospheric model. The responsible parametrization was not designed for the km scale and application at this resolution results in a too early onset of convective precipitation."

*"line 331: the potential (dis)agreement of simulated and actual land cover can be checked using high-resolution land cover datasets such as provided by ESA CCI.*

An explanation was included (lines 386-389):

"To avoid a biased comparison related to land-cover mismatches between the simulation and the actual land use at the observation sites, the simulation results are averaged over five-by-five atmospheric grid boxes centered around the observation sites thus giving approximately the same fractional land cover as is present at the observation location."

*line 343: how are the temperature standard deviations determined?"*

We explained how the temperature deviations are calculated (lines 399-402 and appendix):

"The simulated temperature standard deviations (mean absolute difference for each time of day between the specific daily value and the corresponding monthly mean, see appendix Formula A1 for details) are somewhat smaller than observed, especially for afternoons in the summer half year with underestimations of the temperature standard deviation larger than 20%."

"7.3 Appendix Formulas

$$\sigma_{T,t} = \frac{1}{days} \sum \left| T_{days,t} - \overline{T}_t \right|$$

Equation A1: σ is the temperature standard deviation and the subscript $t$ denotes the time of day. This is calculated separately for each month of the year to create the 12 profiles. The overbar for the temperature $T$ denotes the monthly mean temperature value while the subscript days,t indicates that this is the daily value for the respective time of day."

"*line 361: 'very well', please be more specific and objective*"

Changed wording and added more detail (lines 420-421):

"Overall, the atmospheric profiles, including the ABL heights, are very close to observations during the day and at heights above 10m."

"*line 367: why not using the ESA CCI soil moisture dataset derived from observations of various satellites for this validation?*"

Since we use different land-use and soil properties compared to the observational dataset a comparison would not give any conclusive result. We already acknowledge a large bias in the results and one of similar magnitude can be expected when comparing to a different dataset.

"*lines 394-395: I do not really understand why this daily matching is applied here? Also it is not clear how this is done.*"

This means that for each day a factor is calculated and the actual result is corrected by this factor (lines 452-456):

"With that, a daily matching of the cumulative distribution functions of the simulated catchment and satellite retrieved soil moisture is performed to find a factor which then is assumed to be the soil-moisture bias of the simulation and is applied as a correction factor."

"*lines 391 & 396: I guess you are referring to Figure 15 here, not Figure 16 as stated.*"

Figures have changed greatly, all figures should now be referenced correctly.

"*line 415: 'adequate agreement', please be more specific and objective*"

Changed wording (lines 474-476):

"The range of the hydrological responses to precipitation in the simulated catchment is similar to the observations and also during dry periods the behavior is similar, which is noteworthy since no calibration to runoff data has been applied with the model."

"*line 422: 'will always be replaced' needs to be tuned down in my opinion*"

We now say "often" instead of "always".

*"line 440: 'good distribution', please be more specific and objective"*

Changed lines 500-502:

"First, we visually inspect the groundwater depth map, shown in Figure 13a. Accordingly, the model shows a reasonable split between shallower and deeper (5 meter and below) groundwater tables compared to expected values from observations with shallower levels overall."

*"lines 447-450: I do not understand how the "fluctuations" are "scaled". Do you divide by the inter-annual standard deviation to obtain normalized anomalies (or z-scores)? If so, please name it this way as the term "fluctuations" is rather unclear."*

In this case fluctuation means that the yearly average is removed for both time series (lines 508-512):

"Instead, we compare (1) the magnitude of the fluctuation in the groundwater table throughout the catchment during a year (calculated as the groundwater observation minus its yearly mean, shown in Figure 13b) and (2) the average trend of the groundwater level in the full model domain (calculated after subtracting the mean and scaling the fluctuations to have the same magnitude). This means we are comparing standardized anomalies for the observed and simulated groundwater levels."

*"line 450: I guess this should be "according to Figure 19b" and not 19c?"*

Again, figures have changed, should be correct now.

*"lines 451-452: How is this trend computed?"*

This was poor wording. We changed it to "fluctuations" to indicate that the same time-series as before is referenced (lines 512-514):

"According to Figure 13b, the magnitude of the groundwater fluctuations is within similar ranges as the observations (Figure 13b), while a few observation wells show larger fluctuations."

*"line 454 and following: I like this discussion of limitations and issues"*

Thanks.

*"line 458: to me it seems three challenges being discussed here (?)"*

This was an error left over from a previous version. It is corrected.

*"lines 502-512: As there are multiple concrete ideas to improve the model setup and consequently the dataset, why not implementing them before publishing this dataset? comment if this will be game-changers"*

We have added a comment giving our opinion on this (lines 576-579):

"While these changes would show improvements, they are likely marginal or very specific (river discharge characteristics) and would therefore not warrant the great computational cost to re-run for such a long time. Future developments of TerrSysMP may enable this option and it would be interesting

to compare resulting datasets and quantify the increase of simulation speed by using GPU compute technologies."

*"Figure 2: This figure is the same as in the Gasper et al. 2014 paper, with the reference given in the caption. I think it is uncommon to use figures from previous papers, so I would remove this and only refer to the figure in the reference paper in the main text."*

Moved to Appendix

*"Figure 3: Maybe I missed that but what is domain 1?"*

Corrected, replaced by "simulated domain"

*"Figure 4: "e+00" can be removed"*

removed

*"Figures 4-6: Please label the color bars."*

done

*"Figure 6: Please harmonize "evaporation" and "evapotranspiration" in the caption and the axis label. The same applies for the main text in section 4.1."*

We changed everything to "evaporation" in the text and figures.

*"Figure 8: Values are quite far apart from color bars. Also, it would be nice to also express the difference as percentage."*

We removed the large spaces between values. A percentage difference would show almost the same picture, just with lower values in the mountains. We therefore left this out.

*"Figure 11: y-axis label missing Please explain what is meant with the "temperature standard deviations""*

Added label and explained standard deviations in the text (lines 403-405 and Appendix):

"The simulated temperature standard deviations (mean absolute difference for each time of day between the specific daily value and the corresponding monthly mean, see appendix Formula A1 for details ) are somewhat smaller than observed, especially in afternoons in the summer half year with underestimations of the temperature standard deviation larger than 20%."

"7.3 Appendix Formulas

$$\sigma_{T,t} = \frac{1}{days} \sum \left| T_{days,t} - \overline{T_t} \right|$$

Formula A1: σ is the temperature standard deviation and the subscript t denotes the time of day. This is calculated separately for each month of the year to create the 12 profiles. The overbar of the Temperature T denotes the monthly mean value while the subscript days,t indicates that this is the daily value for the respective time of day."

*"Figure 13: Please specify from which times the reference radiosonde observations are taken. Further, please explain how the standard deviation is derived."*

Same as last figure, also the times are in the figure already. Also added them to the caption.

*"Figure 15: It would be insightful to quantify the agreement of the temporal dynamics with e.g. a correlation."*

This option is discussed in the text instead. We could unfortunately not include it as the processed dataset has been lost, related to a very unusual incident on the supercomputer affecting the supposedly safely stored data, and re-processing from the original data would take a very long time as the person who originally worked on this is no longer involved in the project.

*"Figures 16-18: Please use the station names throughout instead of the position numbers."*

Changed to station names.

*"Figure 19: Panel a is not labelled, as well as color bar and axes therein The terms "model" and "reality" are not consistent with the terminology used throughout the manuscript."*

Changed the figure accordingly.

**Reviewer 2**

*"However, I still am left with the question of what is this data going to be used for beyond the work done by the authors. From my understanding, a publication in ESSD should be a dataset that one could expect to be used extensively by the larger community. If this covered the entire country or Europe then that would be a different story, but I just don't see it as is. I still believe the work should be published but the paper would greatly benefit by putting it all in a greater context. I suggest the discussion provide a much more detailed overview of what this data could be used for and why it is necessary to have a unique dataset instead of just having each researcher rerun the simulations. In essence, it would be nice to know what this dataset could provide the larger scientific community 5-10 years from now."*

The second reviewer raises a similar point as the first one. So we iterated on our initial response and added more detail. In the following we supply this answer once more:

"First, we want to address the concern of a too small domain used. Given the complexity of our model system driven by the challenge to determine the evolution of the water and energy fluxes in the land-atmosphere system as complete as possible with a most advanced model system able to most directly simulate all relevant processes, we are limited by the currently available computational resources. Our wish to create a dataset of several years in order to allow for climatological analyses, we were forced to restrict ourselves spatially. Given the new hard- and software developments such as e.g. GPU computing, such simulations will potentially be possible for Europe probably in about a decade. However – even then – problems will arise on the availability of sufficiently resolved and homogeneous information on the soil and sub-soil required for such simulations. We doubt that the quality of the respective data sets for the German state of Baden-Württemberg, which we used in our work, will be available over Europe in a similar resolution and/or granularity. Under such conditions the results of spatially much more extended simulations would most probably signal more the different data sets than the processes – and thus restrict any analysis to the regions with consistent soil and – sub-soil data.

We see a range of applications for our data sets. Users may repeat our simulations with their own model system or use our system and experiment with alternative or improved input data or parameterizations and evaluate according sensitivities by comparison with our simulations, which would serve as the benchmark. This is currently not possible, because similar almost decade long fully coupled simulations of comparable complexity are not available but will for sure in the future. Since the information provided by our data set does allow users to apply observation operators for pseudo-observations, data assimilation experiments in the coupled system can be performed, which is what we are currently doing.

Since it was our goal to use the highest spatial grid resolutions and to reduce parameterization as much as possible under the computational constraints posed by currently available IT resources, the data set can be used to perform daily, seasonally and long-term analyses of intra- and intercompartmental water and heat energy fluxes and budgets in the land-vegetation-atmosphere system. The allows e.g. for statistical analyses of the relations between the state and evolution of the soil moisture and temperature at any layer and the state of the atmospheric boundary layer and even precipitation

patterns and vice versa taking into account also arbitrary lag-times. Given the diversity of landscape and land use contained in our simulations the dependency of such relations on the latter and on season can also be distilled from the data set. Since we cover a rather long time period and – given the comprehensiveness of the model system – large area, users can restrict such analyses also to sub-areas (e.g. smaller sub-catchments) or interesting time periods, certain times of the day, or seasons. Interesting examples are e.g. the dependency of convection in the atmosphere on the detailed state of the land and vegetation state and its heterogeneity in rather realistic settings or e.g. the magnitude of canopy evaporation after rain events at certain times of the day for a range of wind and temperature conditions.

If re-simulations with other model setups (e.g. different parameterizations or spatial resolution) are performed, sensitivities to model configurations or parameterizations can be determined. This includes the ability to attribute changes such as differences in atmospheric variables (for example 2m air temperature and air humidity) to a changed setup of the land surface. For our ongoing ensemble-based data assimilation experiments, we use due to prohibitive IT resources runs at lower resolutions, which clearly introduced biases especially in the subsurface. Also the impact of even more detailed parametrizations can easily be tested. E.g. one could alternatively use the tiling approach possible in CLM with several PFTs in one grid cell instead of only the dominant one as we did, or just a different microphysics scheme in the atmospheric component. Most probably only limited and well selected time periods would need to be re-run for such analyses.

In the future our setup can be extended to an ensemble. The data provided with our data set can be easily used to produce such an ensemble. E.g. the methods used to generate the soil and sub-soil can be used to produce a set of equally likely soil configurations given the always limited observations while atmospheric variability and uncertainty can be generated by the use of analysis ensembles for initialization and lateral boundary conditions."

We added the following text to the discussion section (lines 588-607):

"Finally, we want to address the applicability and usefulness of this dataset for various studies. As indicated, this dataset can be valuable for data assimilation both for testing new methods or algorithms and as a standard set for synthetic observations to pull from. It is thus possible to carry out data assimilation experiments with different conditioning datasets. Due to the long time series we have covered almost any possible weather regime (with the exception of truly extreme events) which can be a great advantage as some algorithms may work well for most conditions but may show weaknesses for other specific conditions (for instance the CMEM operator in combination with frozen soils). It also allows to investigate the impact of simplifications such as using a fixed atmospheric forcing instead of a model and thus disregarding feedback mechanisms. Next to data assimilation there are also model development and model analysis and comparison studies that can benefit from this dataset. If specific changes to the model system are made, for example testing a new cloud parametrization, all of the input files that are provided with this dataset can be used to quickly set up a working environment with known results to compare to. Here the length of the simulation is again an advantage since any development can be tested for relevant time slices. A detailed analysis of the dataset regarding compartment interactions is also of interest. We have shown the overall behavior of the system but we have not studied specific interesting events in detail. It would also be of interest to perform longer term simulations to analyze climate change and analyze better inter-annual variabilities by considering yearly

changes in the LAI cycle. Lastly, this setup can also be considered as a template for ensemble-based setups in the future. Right now, reduced resolutions are needed in order to run many members of such a coupled model system. As we have shown, even this higher resolved simulation still shows some biases that are directly related to resolution so increasing resolution also in ensemble simulations will be a logical step in the future to obtain better results. When this happens, the methods we used here to generate this simulation will be very useful as well as the analysis presented here to decide how an ensemble should be set up based on the goal (an ensemble for flood forecasts would benefit from a different strategy than an ensemble for drought monitoring)."